# Cell cycle-dependent phosphorylation regulates RECQL4 pathway choice and ubiquitination in DNA double-strand break repair

Huiming Lu [1,2], Raghavendra A. Shamanna[1], Jessica K. de Freitas[1], Mustafa Okur[1], Prabhat Khadka[1], Tomasz Kulikowicz[1], Priscella P. Holland[1], Jane Tian[1], Deborah L. Croteau[1], Anthony J. Davis[2] & Vilhelm A. Bohr[1]

Pathway choice within DNA double-strand break (DSB) repair is a tightly regulated process to maintain genome integrity. RECQL4, deficient in Rothmund-Thomson Syndrome, promotes the two major DSB repair pathways, non-homologous end joining (NHEJ) and homologous recombination (HR). Here we report that RECQL4 promotes and coordinates NHEJ and HR in different cell cycle phases. RECQL4 interacts with Ku70 to promote NHEJ in G1 when overall cyclin-dependent kinase (CDK) activity is low. During S/G2 phases, CDK1 and CDK2 (CDK1/2) phosphorylate RECQL4 on serines 89 and 251, enhancing MRE11/RECQL4 interaction and RECQL4 recruitment to DSBs. After phosphorylation, RECQL4 is ubiquitinated by the DDB1-CUL4A E3 ubiquitin ligase, which facilitates its accumulation at DSBs. Phosphorylation of RECQL4 stimulates its helicase activity, promotes DNA end resection, increases HR and cell survival after ionizing radiation, and prevents cellular senescence. Collectively, we propose that RECQL4 modulates the pathway choice of NHEJ and HR in a cell cycle-dependent manner.

[1] Laboratory of Molecular Gerontology, National Institute on Aging, National Institutes of Health, Baltimore, MD 21224, USA. [2] Division of Molecular Radiation Biology, Department of Radiation Oncology, UT Southwestern Medical Center, Dallas, TX 75390, USA. Correspondence and requests for materials should be addressed to V.A.B. (email: vbohr@nih.gov)

DNA double-strand breaks (DSBs) are generated by endogenous stress, during programmed recombination events, or after exposure to exogenous sources, such as ionizing radiation (IR) and chemotherapeutics[1]. DSBs are highly cytotoxic DNA lesions, and the rate of spontaneous production of endogenous DSBs in a cell is approximately 50 per cell cycle[2]. Deficient DSB repair leads to genome instability, chromosomal rearrangements, cellular senescence, and cell death[3,4]. Cells primarily utilize two major pathways, non-homologous end joining (NHEJ) and homologous recombination (HR), to repair DSBs[4]. NHEJ initiates with the Ku70-Ku80 heterodimer (Ku) binding to the DNA ends. It then recruits the NHEJ machinery to the DSB to process and religate the DSB independent of a DNA template[4,5]. HR uses a homologous DNA sequence as a template to mediate repair of a DSB. It starts with binding of the MRE11–RAD50–NBS1 complex (MRN) to the DSB, where in conjunction with CtIP they initiate short DNA end resection. Extensive resection of the 5′ end of DSB is then performed by the DNA2-BLM or EXO1 pathway, which creates 3′ protruding single-strand DNA (ssDNA) tails[3,4,6]. The ssDNA tails are bound and protected by RPA, which is then replaced by RAD51. Next, the RAD51-bound ssDNA tails perform strand exchange with intact sister chromatids, followed by DNA synthesis, and HR then is completed by resolution or dissolution of the Holliday junctions[7].

DSB repair pathway choice is a tightly regulated process that is influenced by many factors, including the cell cycle phase, DNA end resection, and post-translational modifications[4]. The pathway choice between NHEJ and HR is coordinated throughout the cell cycle[4,8]. NHEJ can function in all phases of the cell cycle, but is most active in G1, whereas HR is highly active in the S and G2 phases, when sister chromatids are available as substrates for recombination. Cyclin-dependent kinases (CDKs), with their cyclin partners, drive cell cycle progression[9]. The overall activity of CDKs is low in G1 and increases progressively during the cell cycle after the G1/S boundary[9]. CDKs phosphorylate many DNA repair proteins to facilitate DNA resection[10–14], which is a fundamental process for cell cycle-dependent regulation of DSB repair pathway choice[15]. In addition to CDK-mediated phosphorylation, ubiquitin-dependent regulation also operates at multiple levels to contribute to DSB repair pathway choice in a cell cycle-dependent manner[16–19]. For example, the DDB1-CUL4

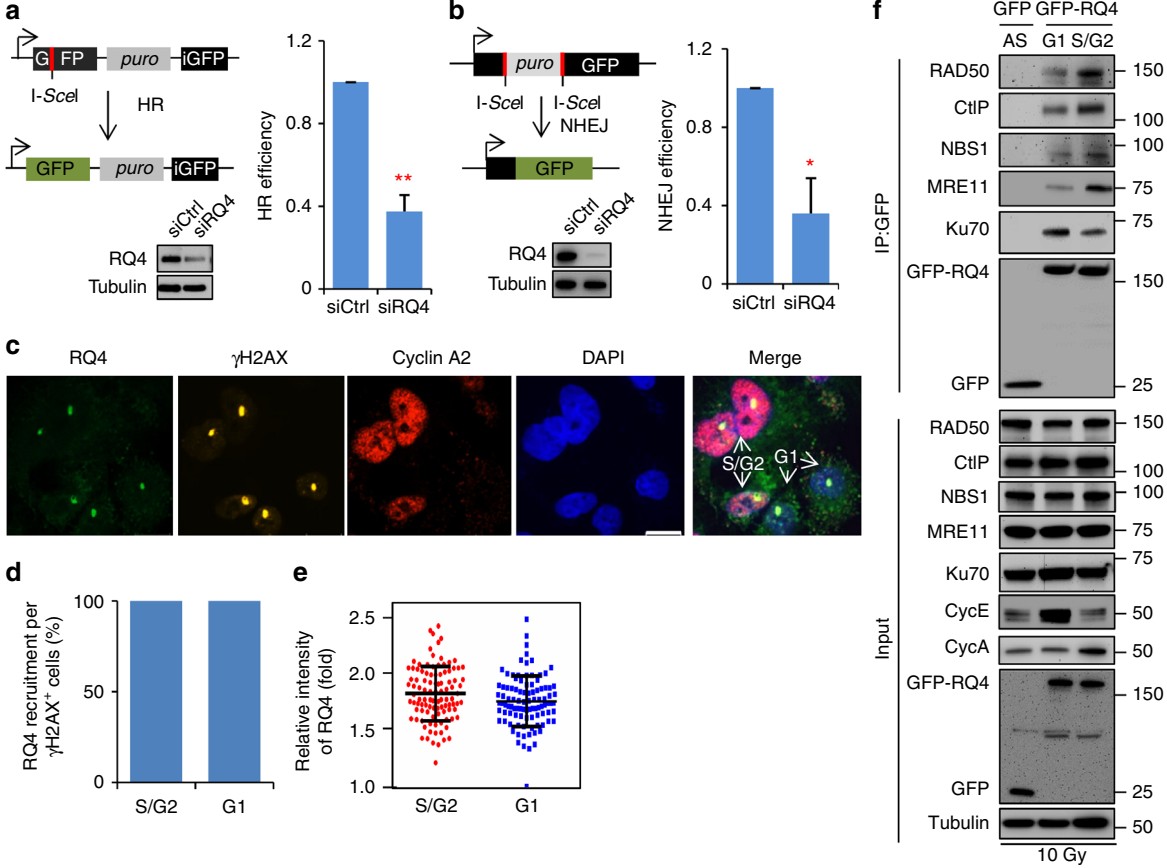

**Fig. 1** The interaction of RECQL4 with MRE11 and Ku70 is regulated during cell cycle to promote HR and NHEJ. **a** Depletion of RECQL4 reduces HR in DR-GFP U2OS cells. **b** Knockdown of RECQL4 decreases NHEJ in EJ5 U2OS cells. Three days after transfection of *RECQL4* siRNA into U2OS DR-GFP or EJ5 U2OS cells, I-SceI endonuclease was expressed to induce DSBs in GFP reporters for both HR and NHEJ repair assays. GFP-positive cells were examined by flow cytometry 3 days later. The results were shown mean ± s.e.m. from three biological repeats, with *p*-values determined by Student *t*-tests. *$p<0.05$; **$p<0.01$. **c**–**e** Recruitment of endogenous RECQL4 to micro-point laser-induced DSBs in U2OS cells. The wavelength of micro-point laser is 435 nm. RECQL4, γH2AX, and Cyclin A2 were visualized by immunostaining with the relevant antibodies. DSB tracks and cells in S/G2 phase were indicated by γH2AX and Cyclin A2, respectively. About 100 cells were quantified for each, for Cyclin A2 positive, $n=102$; for Cyclin A2 negative, $n=98$. **c** Representative image; scale bar, 17 μm. **d** Percentage of cells with RECQL4 recruitment in γH2AX-positive cells. **e** Relative intensity of RECQL4 at DSB tracks. **f** RECQL4 interacts with MRN/CtIP and Ku70 in S/G2 and G1 phases, respectively. HEK293T cells expressing GFP-tagged RECQL4 were synchronized in G1 phase of the cell cycle by 16-h treatment of 2 μg mL$^{-1}$ aphidicolin. To enrich for S/G2 cells, the synchronized cells were then released into fresh medium for 10 h. The cell cycle distribution of the cells was analyzed with PI staining and also protein level of Cyclin A and Cyclin E. The cells were irradiated with γ rays for 10 Gy before subjected to GFP Immunoprecipitation

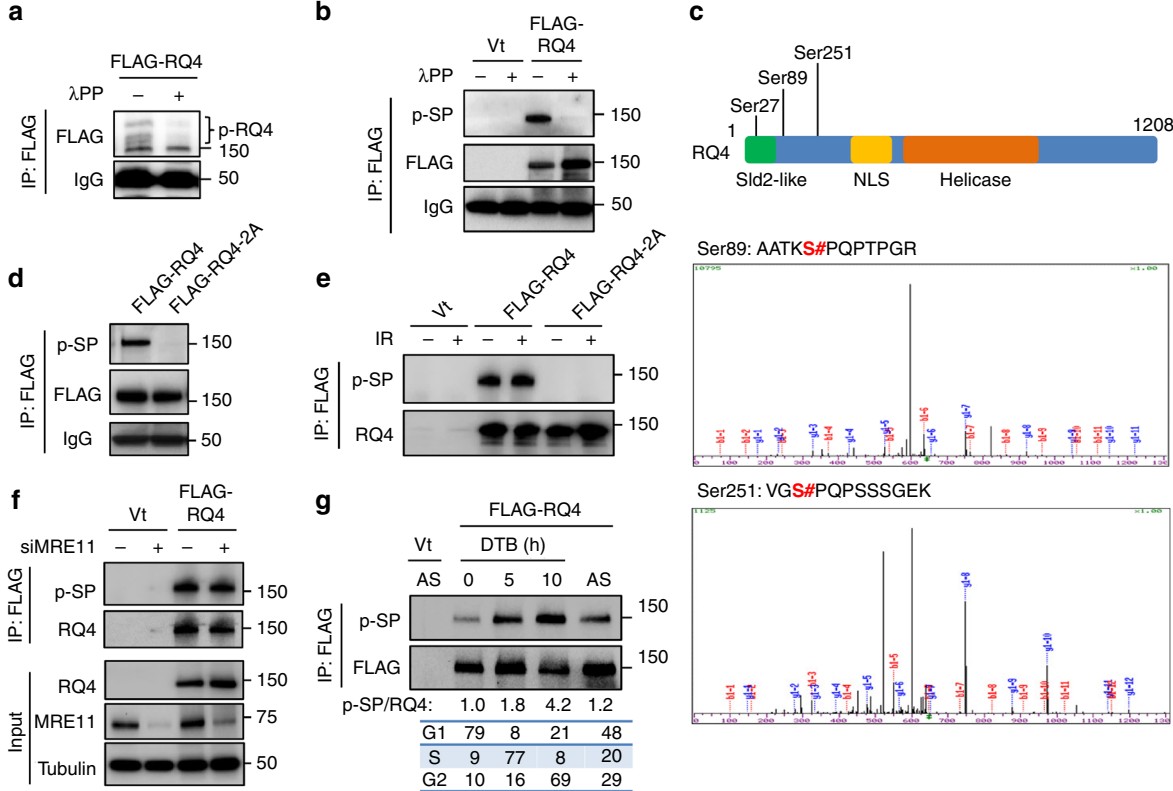

**Fig. 2** CDK-mediated phosphorylation of RECQL4 occurs in S/G2 phases of the cell cycle. **a** RECQL4 is phosphorylated in HEK293T cells. 3×FLAG-tagged RECQL4 was purified from HEK293T cells, and half was treated with λPP. The proteins were separated by SDS-PAGE with Phospho-tag™ Acrylamide. Slower migrating RECQL4 bands indicate phosphorylation. **b** RECQL4 is phosphorylated by CDKs in HEK293T cells. Phosphorylation on serine of CDK-targeting SP motifs (p-SP) was detected on 3×FLAG-tagged RECQL4. **c** Identification of phosphorylation sites Ser89 and Ser251 on RECQL4 by mass spectrometry. Upper panel shows schematic diagram of RECQL4. Lower panels are spectra of identified peptides containing phosphorylated Ser89/Ser251. Each site is followed by a proline to form a CDK-targeting SP motif. The annotation of all three spectra is given in Supplementary Fig. 1. **d** Western blotting analysis of p-SP level on 3×FLAG-tagged RECQL4 and RQ4-2A mutant (Ser89Ala/Ser251Ala), which were purified from HEK293T cells. **e** IR treatment does not alter p-SP level of RECQL4. Wild-type RECQL4 and RQ4-2A were purified from untreated or irradiated HEK293T cells and p-SP were analyzed by Western blotting. **f** Depletion of MRE11 does not change phosphorylation of SP sites in RECQL4 in HEK293T cells. 3×FLAG-tagged RECQL4 was purified from siMRE11 or siCtrl-treated HEK293T cells, and p-SP was measured through Western blotting. **g** Increase of phosphorylation of SP sites in RECQL4 in the S and G2 phases of the cell cycle in U2OS cells. U2OS cells were synchronized in the G1 phase with double thymidine block (DTB), were released in fresh medium for 5 and 10 h to enrich S phase and G2 phase cells, respectively. The plasmid expressing 3×FLAG-RECQL4 was transfected into U2OS cells during the first release as showed in Supplementary Fig. 2a. Cell cycle distribution was analyzed with PI staining and protein levels of Cyclin A and Cyclin E. Phosphorylation of SP sites in 3×FLAG-tagged RECQL4 was measured by Western blotting. NLS nuclear location sequence

E3 ubiquitin ligase, a member of the cullin-RING E3 ubiquitin ligase family, regulates cell cycle progression, DNA replication, and genome integrity[20]. Both core components, DDB1 and CUL4A, are recruited to laser-induced DSBs, and the DDB1-CUL4 E3 ubiquitin ligase promotes DNA end resection and HR by regulating histone protein monoubiquitination[21,22]. Proper coordination of DSB repair pathway choice plays a significant role in maintaining genome stability[23]. However, many questions remain about understanding the underlying mechanisms[4].

RECQL4, one of the five RecQ helicases in mammalian cells, guards the genome by promoting DNA replication, DNA repair, and telomere maintenance[24]. It is also tightly linked with prevention of aging and cancer[25,26]. Defects in human RECQL4 are associated with three autosomal disorders: Rothmund-Thomson Syndrome (RTS), RAPADILINO syndrome, and Baller-Gerold Syndrome[26,27]. The dysfunction of RECQL4 leads to increased cellular senescence and apoptosis due to the accumulation of DNA damage, which contributes to the clinical features observed in RTS mouse models[28–30]. Fibroblasts from RTS patients and RECQL4-depleted cells are sensitive to IR and RECQL4 rapidly accumulates at DSBs, suggesting that RECQL4 plays a role in DSB repair[31,32]. We previously reported that RECQL4 promotes NHEJ

by interacting with the Ku complex[33]. Moreover, we recently identified an important role for RECQL4 in promoting HR[34]. RECQL4 forms a complex with the DNA resection proteins MRN and CtIP, promoting DNA end resection[34]. As RECQL4 promotes both NHEJ and HR, we investigated the mechanism of this coordination. Since NHEJ is active throughout the cell cycle and HR occurs primarily in S/G2 phases, we hypothesized that an S/G2-specific post-translational modification of RECQL4 could affect its role in pathway choice. Here, we report a cell cycle-dependent regulation of RECQL4 in DSB repair pathway choice, driven by its CDK1/2-mediated phosphorylation and DDB1-CUL4A-dependent ubiquitination in S/G2 phases of the cell cycle.

## Results

**Cell cycle-regulated interaction of RECQL4 with Ku and MRN.**
Initially, we wanted to verify that RECQL4 promotes both HR and NHEJ. To measure HR and NHEJ, we used the reporter cell lines DR-GFP U2OS[35] and EJ5 U2OS[36]. In DR-GFP U2OS cells, I-SceI endonuclease generates a DSB in the cassette of the dysfunctional GFP gene, which can be repaired via HR using the

downstream GFP fragment (Fig. 1a)[35]. Knockdown of RECQL4 protein decreased HR efficiency by over 60% (Fig. 1a), which is similar to our previous findings[34]. In EJ5 U2OS cells, I-SceI-induced DSBs are repaired by NHEJ to restore a functional GFP

gene (Fig. 1b)[36]. Depletion of RECQL4 also significantly reduced NHEJ efficiency by over 60% in EJ5 U2OS cells (Fig. 1b). Collectively, the results support that RECQL4 promotes DSB repair mediated by both HR and NHEJ.

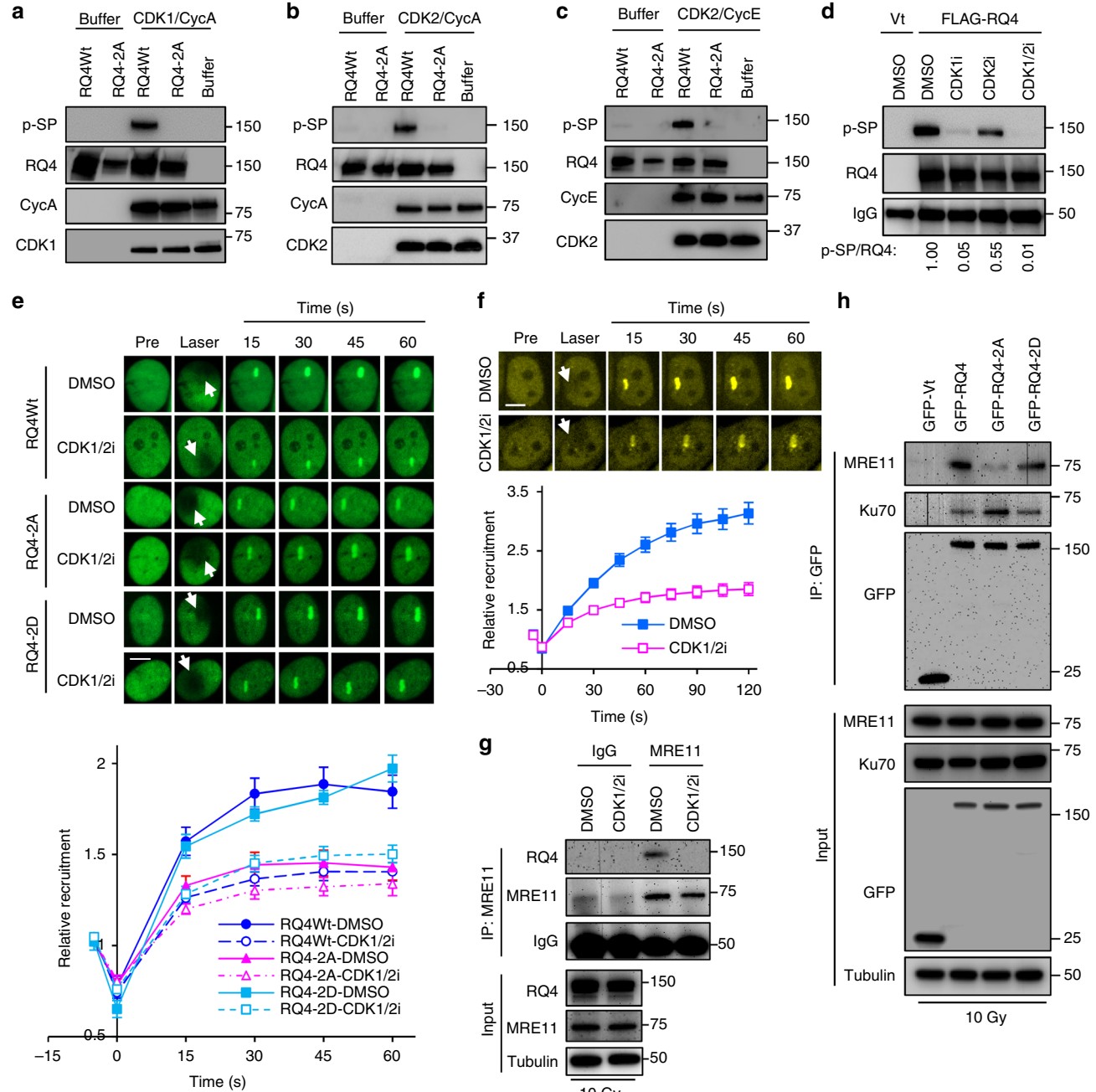

**Fig. 3** CDK1/2-dependent phosphorylation promotes RECQL4 participation in repair of DSBs. **a–c** In vitro phosphorylation of RECQL4 by CDK1/Cyclin A (**a**), CDK2/Cyclin A (**b**), and CDK2/Cyclin E (**c**). 3×FLAG-tagged RECQL4 WT and RQ4-2A mutant were purified from HEK293T cells, and pre-treated with λPP. After washing off λPP, the RECQL4 proteins were incubated with CDK1/2 and their cyclin partners. Phosphorylation of RECQL4 SP sites were analyzed by Western blotting. **d** CDK1/2 inhibitors repress p-SP level of RECQL4. HEK293T cells expressing 3×FLAG-RECQL4 were pretreated with 10 μM RO3306 (CDK1i), 10 μM CDK2i-III (CDK2i) or combination of CDK1i and CDK2i for 4 h. Then, 3×FLAG-tagged RECQL4 were purified and SP sites' phosphorylation was analyzed by Western blotting. **e** Ser89/Ser251 phosphorylation promotes recruitment of RECQL4 to laser-induced DSBs. GFP-tagged RECQL4 Wt, RQ4-2A, and RQ4-2D were expressed in U2OS cells, and the cells were stripped with micro-point laser after treatment with CDK1 and CDK2 inhibitors or DMSO. The relative intensities of these cells are presented as mean ± s.e.m. The number of cells were quantified, for RQ4Wt, n = 9; RQ4Wt with CDK1/2i, n = 16; RQ4-2A, n = 19; RQ4-2A with CDK1/2i, n = 17; RQ4-2D, n = 26. RQ4-2D with CDK1/2i, n = 19. Scale bar, 5 μm. **f** Inhibition of CDK1 and CDK2 attenuated recruitment of YFP-MRE11 to laser-induced DSBs. For DMSO, n = 27; for CDK1/2i, n = 49. **g** Inhibition of CDK1 and CDK2 represses RECQL4's interaction with MRE11. The U2OS cells pretreated with CDK1 and CDK2 inhibitors or DMSO were irradiated for 10 Gy and processed for IP with anti-MRE11 antibody 5 min after irradiation. **h** Ser89/Ser251 phosphorylation promotes the interaction of RECQL4 with MRE11. GFP-tagged RECQL4, RQ4-2A, and RQ4-2D were immunoprecipitated from HEK293T cells after a 10 Gy IR treatment. MRE11 and Ku70 were measured in the IP products

RECQL4 is recruited to DSBs[32], but it is unknown whether this recruitment depends on the cell cycle. Thus, we immunostained endogenous RECQL4 in U2OS cells after micro-point laser irradiation, using cyclin A2 and γH2AX as makers for cells in S/G2 phase and for DSBs, respectively[13,37]. RECQL4 co-localized with γH2AX in all laser-damaged cells, independent of the cell cycle (Fig. 1c, d). Moreover, the intensity of RECQL4 recruitment in S/G2 cells did not differ from that in G1 cells (Fig. 1c, e). The results clearly show that RECQL4 recruitment to DSBs is similar throughout the G1, S, and G2 phases of the cell cycle.

Previously, we showed that RECQL4 interacts with the Ku heterodimer, the MRN complex, and CtIP, which are required to initiate NHEJ and HR, respectively[33,34]. To test our hypothesis that RECQL4 promotes NHEJ and HR in a cell cycle-dependent manner, we asked whether RECQL4 preferentially interacts with Ku and MRN/CtIP in G1 and S/G2 phases, respectively. GFP-RECQL4 and its interacting proteins were immunoprecipitated (IPs) from HEK293T cells after aphidicolin induced cell cycle synchronization. The cells were irradiated with 10 Gy to induce DSBs. RECQL4 showed a higher affinity for Ku70 in G1 cells than in S/G2 cells (Fig. 1f). In contrast, the RECQL4 interaction with MRN and CtIP was significantly higher in S/G2 cells than in G1 cells (Fig. 1f). Thus, RECQL4 preferentially interacts with Ku in G1 phase and with MRN/CtIP in S/G2 phase of the cell cycle, and suggests that RECQL4 promotes NHEJ and HR in different cell cycle phases.

**RECQL4 is phosphorylated on Ser89 and Ser251 in S/G2 phases**. Since RECQL4 interacts with Ku and MRN/CtIP in a cell cycle-regulated manner, we postulated that RECQL4 might be a candidate for phosphorylation by CDKs. We IP 3×FLAG-RECQL4 from HEK293T cells and detected different migration patterns using a SDS-PAGE containing phospho-tag[38], which notably decreased after treatment with *E. coli* lambda phosphatase (λPP) (Fig. 2a), indicating that RECQL4 is phosphorylated in vivo. CDKs are members of the proline-directed kinase family that exclusively phosphorylate serines (Ser) or threonines immediately preceding a proline (SP/TP motifs)[39]. To test whether RECQL4 is a substrate of CDKs, phosphorylated Ser in SP motifs (p-SP) were examined using an anti-p-SP antibody[13]. The purified 3×FLAG-RECQL4 showed strong immunoactivity with anti-p-SP antibody, which was abolished by λPP treatment (Fig. 2b), suggesting that RECQL4 is phosphorylated on SP motifs.

To map the phosphorylation sites in RECQL4, 3×FLAG-RECQL4 was purified from HEK293T cells (Supplementary Fig. 1a), and analyzed by comprehensive mass spectrometry. Three phosphorylation sites were identified, Ser27, Ser89, and Ser251 (Fig. 2c and Supplementary Fig. 1b, c). Ser27 is followed by a glutamine to form a SQ motif, a typical substrate for ATM, ATR, or DNA-PK, while Ser89 and Ser251 are followed by prolines to form SP motifs, and these sites are conserved in mammals (Supplementary Fig. 1d). To confirm the mass spectrometry data, both Ser89 and Ser251 were replaced with alanine in RECQL4, designated as RQ4-2A, and the p-SP levels were examined. Wild-type (WT) RECQL4 had a high level of p-SP immunoreactivity, while RQ4-2A was barely detectable with the anti-p-SP antibody (Fig. 2d), indicating that Ser89 and Ser251 are major targets for CDK-mediated phosphorylation in RECQL4.

Next, we investigated whether phosphorylation of SP sites in RECQL4 is induced by DNA damage. The p-SP levels in WT RECQL4 and RQ4-2A were not changed after IR treatment (Fig. 2e). MRE11 directly interacts with CDK2 and regulates CtIP phosphorylation[40], so we tested its role in CDK-mediated

phosphorylation of RECQL4. Depletion of MRE11 did not alter the p-SP level in RECQL4 (Fig. 2f), although RECQL4 interacts with MRE11 as shown above. Activity of CDKs changes with cell cycle progression[8]. Thus, we investigated the phosphorylation status of SP sites in RECQL4 during the cell cycle. With double thymidine block (DTB) treatment, approximately 80% of the U2OS cells were arrested in the G1 phase of the cell cycle, with high levels of Cyclin E and low levels of Cyclin A (Supplementary Fig. 2a, b), and phosphorylation of RECQL4 on the SP sites was low (Fig. 2g). When G1-arrested U2OS cells were released into S and G2 phases of the cell cycle, phosphorylation of RECQL4 increased 1.8-fold and 4.2-fold, respectively (Fig. 2g). In addition, HEK293T cells, arrested in G1 by aphidicolin treatment, also showed lower levels of phosphorylation of SP sites in RECQL4, and this increased markedly when cells were released into S and G2 (Supplementary Fig. 2c–e). These results demonstrate CDK-mediated phosphorylation of RECQL4 on Ser89/Ser251 in the S/G2 phases of the cell cycle.

**CDK1 and CDK2 promote recruitment of RECQL4 to DSBs**. To identify RECQL4-targeting CDKs, RECQL4-interacting proteins were IP with GFP-RECQL4 from HEK293T cells and analyzed by mass spectrometry. CDK1, CDK4, CDK9, and CDK2 were identified by at least three peptides each (Supplementary Fig. 3a and Supplementary Data 1). As shown above, RECQL4 phosphorylation is increased in the S/G2 phases of the cell cycle, suggesting that S/G2-CDKs may affect the increase of p-SP in RECQL4. CDK1/Cyclin A, CDK2/Cyclin E, and CDK2/Cyclin A are all active in the S/G2 phases[9]. Thus, we tested whether RECQL4 could be phosphorylated by CDK1 and CDK2 in vitro. 3×FLAG-tagged WT RECQL4 and RQ4-2A were purified from HEK293T cells and pretreated with λPP to remove background phosphorylation of RECQL4. Subsequently, in vitro phosphorylation reactions were carried out by incubating dephosphorylated RECQL4 with recombinant CDKs and Cyclins. CDK1/Cyclin A phosphorylated WT RECQL4 but not RQ4-2A proteins (Fig. 3a), and CDK2 also phosphorylated WT RECQL4 with Cyclin A (Fig. 3b) or Cyclin E (Fig. 3c), suggesting physical and functional interactions between RECQL4 and CDK1/2.

To further explore whether RECQL4 is targeted by CDK1/2 in vivo, HEK293T cells were treated with selective inhibitors RO3306 (CDK1i) and CDK2i-III (CDK2i). CDK1i and CDK2i treatment inhibited phosphorylation on SP sites in RECQL4 by 95 and 45%, respectively (Fig. 3d), and a combination of CDK1i and CDK2i completely abolished SP phosphorylation of RECQL4 (Fig. 3d). In addition, we tested CDK-dependent phosphorylation of endogenous RECQL4 after roscovitine treatment. Roscovitine is a pan-CDK inhibitor for CDK1, CDK2, CDK5, CDK7, and CDK9[41]. This resulted in a reduction of RECQL4 phosphorylation by 68% (Supplementary Fig. 3b). The CDK2 inhibitor CDK2i-III also repressed RECQL4 phosphorylation on SP motifs in U2OS cells (Supplementary Fig. 3b). Collectively, we find that CDK1 and CDK2 target Ser89/Ser251 of RECQL4 for phosphorylation.

Since RECQL4 is a substrate for CDK1 and CDK2, we mapped the interaction regions of RECQL4 with CDK1 and CDK2. RECQL4 consists of a N-terminal domain containing a Sld2-like motif, a conserved RecQ helicase domain, and a poorly characterized C-terminal domain (Fig. 2c). 3×FLAG-tagged RECQL4 and truncated RECQL4 mutants NT (1-427AA) and RQ/CT (427-1208AA)[42], fused with SV40 nuclear location signal sequence, were IP with their interacting proteins from HEK293T cells. All constructs interacted with CDK1, however, the truncated RECQL4 without the N-terminal domain showed the strongest interaction (Supplementary Fig. 3c). In contrast,

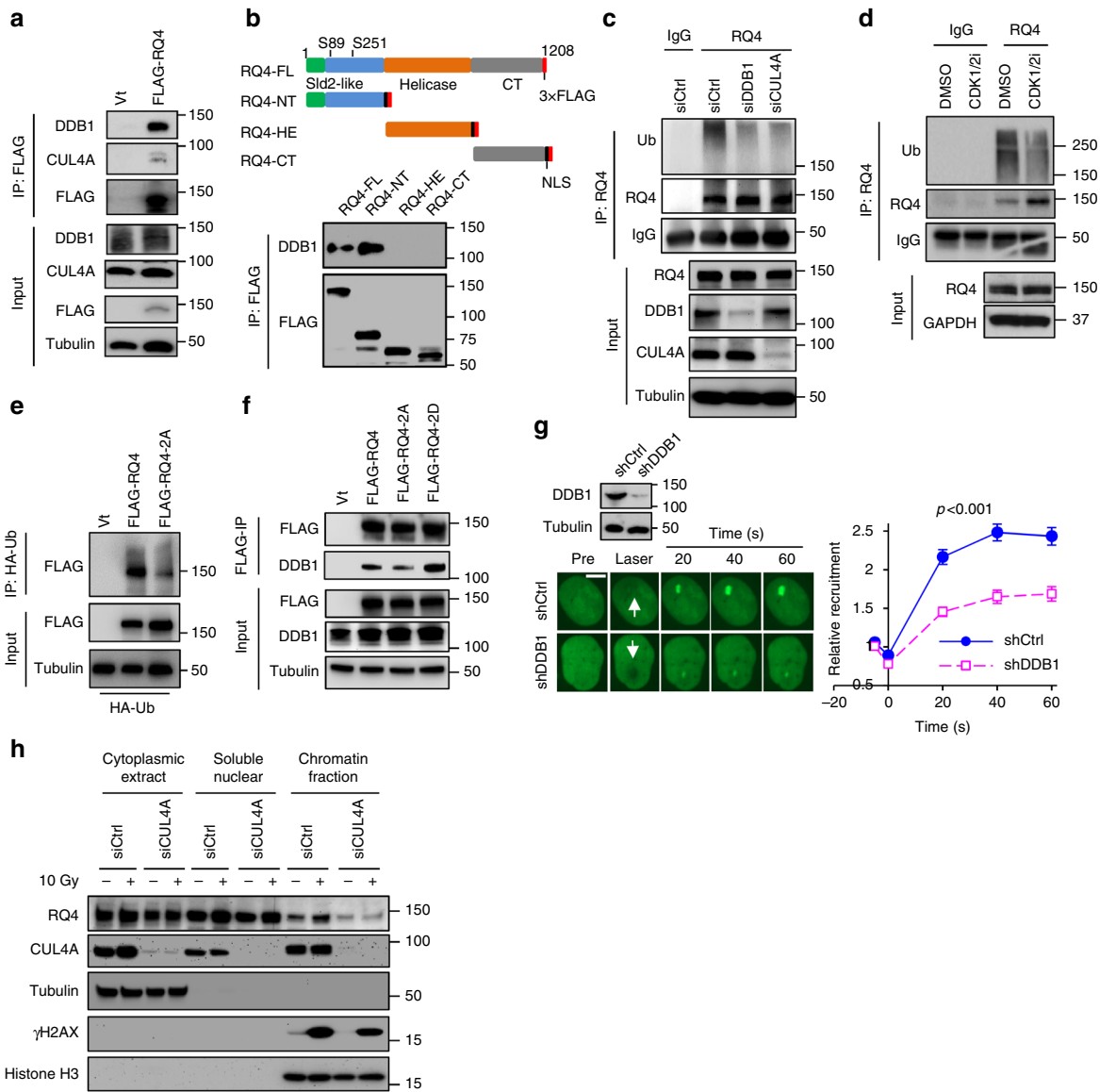

**Fig. 4** DDB1-CUL4A E3 ubiquitin ligase stimulates RECQL4 recruitment to DSBs. **a** Co-IP analysis for the interaction of RECQL4 with DDB1 and CUL4A in HEK293T cells. DDB1 and CUL4A were pulled down with 3×FLAG-tagged RECQL4 from HEK293T cells. **b** N-terminal domain of RECQL4 interacts with DDB1 in HEK293T cells. Upper panel shows schematic diagram of 3×FLAG-tagged RECQL4 and the truncated fragments. N-terminal domain (RQ4-NT), helicase domain (RQ4-HE), and C-terminal domain (RQ4-CT) were tagged with both 3×FLAG and SV40 nuclear-location sequence (NLS). **c** Knockdown of either DDB1 or CUL4A inhibits ubiquitination of RECQL4 in U2OS cells. Endogenous RECQL4 was purified with anti-RECQL4 antibody from denatured U2OS cell lysates and the ubiquitination was then detected with anti-Ub antibody. **d** Inhibition of CDK1 and CDK2 reduced the ubiquitination of RECQL4 in U2OS cells. RECQL4 was immunoprecipitated with anti-RECQL4 antibody from U2OS cells, and its ubiquitination was examined with anti-Ub antibody. The U2OS cells were treated with DMSO or a combination of 10 μM RO3306 and 10 μM CDK2i-III for 4 h before IP. **e** Ser89/Ser251 phosphorylation promotes ubiquitination of RECQL4. HA-tagged ubiquitin and 3×FLAG-tagged WT RECQL4 or RQ4-2A were co-transfected to HEK293T cells, and ubiquitinated proteins were pulled down by HA-IP. Ubiquitinated RECQL4 proteins in the IP product were detected by Western blotting with anti-FLAG antibody. **f** RQ4-2A showed lower affinity with DDB1 in HEK293T cells. 3×FLAG-tagged WT RECQL4, RQ4-2A, and RQ4-2D were pulled down from HEK293T cells and DDB1 protein was probed by Western blotting analysis. **g** Knockdown of DDB1 decreases recruitment of RECQL4 to laser-induced DSBs. DDB1 levels in control and DDB1-depleted U2OS cells were determined by Western blotting. DSBs were generated by micro-point laser. Real-time recruitment of GFP-RECQL4 was observed and quantified. Relative intensity of GFP-RECQL4 is shown as mean ± s.e.m. with *p*-value by Student's *t*-test. The number of cells analyzed: for control cells, *n* = 17; shDDB1, *n* = 24. Scale bar, 5 μm. **h** Depletion of CUL4A in U2OS cells caused failure of RECQL4 recruitment at chromatin after IR stress. Subcellular fractionations of control or *CUL4A* siRNA-treated U2OS cells were prepared and indicated proteins were examined with Western blotting in these fractions

full-length RECQL4 and its N-terminal domain both pulled down CDK2, but the truncated mutant without the N-terminal domain did not (Supplementary Fig. 3d), suggesting that CDK2 preferentially interacts with the N-terminal domain of RECQL4.

To determine the cellular function of the CDK-dependent phosphorylation on RECQL4, we investigated the recruitment of

RECQL4 to DSBs in the presence of CDK1 and CDK2 inhibitors. In agreement with our previous report[34], GFP-RECQL4 was rapidly recruited to laser-induced DSBs in DMSO-treated U2OS cells (Supplementary Fig. 3e). Treatment with CDK1i or CDK2i significantly decreased the accumulation of RECQL4 at DSBs (Supplementary Fig. 3e). Consistent with these findings in U2OS

cells, recruitment of YFP-RECQL4 was also dramatically inhibited in HeLa cells after roscovitine treatment (Supplementary Fig. 3f). In accordance with this finding, treatment with the CDK1 and CDK2 inhibitors also reduced accumulation of endogenous RECQL4 at chromatin after IR (Supplementary Fig. 3g). In contrast, inhibition of ATM with the selective inhibitor KU55933 did not affect RECQL4 recruitment to DSBs (Supplementary Fig. 3h), which is consistent with the previous finding that RECQL4 recruitment in ATM-deficient cells does not differ from that in the normal fibroblasts[32]. These data together demonstrate that CDK1 and CDK2 promote RECQL4's recruitment to DSBs.

Next, we investigated the consequence of blocking RECQL4 phosphorylation by substituting its Ser89 and Ser251 positions with alanines, and found that this attenuated its (RQ4-2A) recruitment to DSBs (Fig. 3e), whereas disruption of a SQ motif by the substitution of Ser27 with alanine did not affect this recruitment (Supplementary Fig. 3i). Moreover, the phosphorylation-mimetic mutant RQ4-2D, containing Ser89 and Ser251 substituted with aspartic acid, showed a similar recruitment pattern as WT RECQL4 (Fig. 3e), suggesting that phosphorylation on Ser89/Ser251 promotes RECQL4 recruitment to DSBs. Interestingly, CDK1 and CDK2 inhibition attenuates recruitment of RQ4-2D to DSBs (Fig. 3e), which led us to examine if an upstream factor important for RECQL4's recruitment to DSBs was affected by CDK inhibition. Since MRE11 promotes RECQL4's recruitment to DSBs[34], we monitored the recruitment dynamics of MRE11 in the absence or presence of CDK1 and CDK2 inhibitors. MRE11's recruitment to DSBs was dramatically attenuated in the presence of CDK1 and CDK2 inhibitors (Fig. 3f). CDK1 and CDK2 inhibitors also blocked accumulation of endogenous MRE11 at chromatin after IR in U2OS cells (Supplementary Fig. 3g). Next, we sought to evaluate the interactions of RECQL4 with MRE11 after CDK1 and CDK2 inhibition. Consistently, CDK1 and CDK2 inhibition repressed the interaction between endogenous MRE11 and RECQL4 in irradiated U2OS cells (Fig. 3g). These results suggest that CDK1 and CDK2 regulate recruitment of MRE11 to DSBs, which explains how CDK1 and CDK2 inhibition affects the recruitment of RQ4-2D to DSBs.

Given that RECQL4's interaction with Ku and MRN is cell cycle-regulated, and that both Ku and MRE11 interact with the N-terminus of RECQL4[33,34], which include Ser89/Ser251, we postulated that Ser89/Ser251 phosphorylation regulates RECQL4's choice of binding partner. We compared the ability of Ku and MRE11 to interact with RQ4-2A and RQ4-2D. RQ4-2D interacted with MRE11, but this was disrupted when the phosphorylation sites were ablated (Fig. 3h). In contrast, Ku preferentially interacted with the RQ4-2A protein compared to the RQ4-2D mutant (Fig. 3h). It should be mentioned that Ku and MRE11 interacted with the WT RECQL4 protein, which was IP from an asynchronous cell population and likely contained a mixture of unphosphorylated and phosphorylated RECQL4. RQ4-2D is recruited to the chromatin fraction in G1 cells after IR stress (Supplementary Fig. 3j), suggesting that phosphorylation of RECQL4 does not affect the interaction with Ku, supported by the data showing that Ku interacts with RQ4-2D (Fig. 3h). These data demonstrate that Ser89/Ser251 phosphorylation regulates RECQL4's interaction with MRE11 and Ku.

**DDB1-CUL4A ubiquitin ligase facilitates RECQL4 in DSB repair**. In addition to CDKs, the mass spectrometry analysis of RECQL4's IP also identified components from the DDB1−CUL4 E3 ubiquitin ligase complex (Supplementary Fig. 4a and Supplementary Data 1). In this complex, CUL4 functions as a scaffold

protein, and its N-terminus and C-terminus interact with linker protein DDB1 and RING finger protein ROC1, respectively[20]. The linker protein DDB1 interacts with the WD40 domain-containing substrate receptor proteins or substrates[20]. Remarkably, 49 peptides of DDB1 were identified, and peptides from other proteins in the complex were also identified including CUL4A and eight WD40 domain-containing proteins (Supplementary Fig. 4a), which are predicted substrate receptors for DDB1-CUL4 E3 ubiquitin ligase[43]. We next performed an IP experiment to confirm the proteomic results, and found that endogenous DDB1 and CUL4A efficiently co-immunoprecipitated with 3×FLAG-RECQL4 (Fig. 4a). By mapping, we found that the N-terminal domain of RECQL4 associates with endogenous DDB1 (Fig. 4b).

The DDB1-CUL4 E3 ubiquitin ligase regulates the cell cycle and genome stability by ubiquitinating key proteins in these processes[44]. To explore whether RECQL4 is ubiquitinated by this E3 ubiquitin ligase, we purified endogenous RECQL4 from U2OS cells after treatment with control siRNA or siRNA against DDB1 or CUL4A. RECQL4 from control cells showed strong ubiquitination, and disruption of DDB1-CUL4A E3 ubiquitin ligase, by depletion of either DDB1 or CUL4A, greatly reduced the ubiquitination level of RECQL4 (Fig. 4c). In addition, we co-expressed HA-tagged ubiquitin and 3×FLAG-RECQL4 in shCtrl or shDDB1-treated HEK293T cells, and IP ubiquitinated proteins from denatured cell lysates. 3×FLAG-RECQL4 was detected with anti-FLAG antibody among the ubiquitinated proteins from shCtrl-treated cells, while ubiquitination of 3×FLAG-RECQL4 decreased dramatically when DDB1 was depleted (Supplementary Fig. 4b). Together, these data provide strong evidence that RECQL4 is ubiquitinated by the DDB1-CUL4A E3 ubiquitin ligase.

Phosphorylation and ubiquitination play cooperative roles in the DNA damage response cascade[45,46]. Therefore, we examined whether there is coordination between CDK1/2-mediated phosphorylation and DDB1-CUL4A-dependent ubiquitination of RECQL4. Treating U2OS with CDK1i and CDK2i resulted in a reduction in ubiquitination of endogenous RECQL4 (Fig. 4d). Consistently, the ubiquitination of the phosphorylation-null mutant RQ4-2A showed less ubiquitination than WT RECQL4 (Fig. 4e). Conversely, depletion of DDB1 did not change the p-SP levels of WT RECQL4 (Supplementary Fig. 4c). Moreover, RQL-2A showed less affinity to DDB1, whereas RQ4-2D pulled down more DDB1 (Fig. 4f), indicating that Ser89/Ser251 phosphorylation enhances the interaction between RECQL4 and DDB1. Collectively, these data suggest that CDK1/2-mediated phosphorylation of RECQL4 promotes its ubiquitination by DDB1-CUL4A E3 ubiquitin ligase.

Since the Ddb1−Cul4 E3 ligase complex promotes recruitment of Exo1 and facilitates Exo1-mediated DNA resection and HR in yeast[22], we quantified the recruitment of GFP-RECQL4 in DDB1-depleted and control U2OS cells. Knockdown of DDB1 greatly decreased GFP-RECQL4 recruitment to laser-induced DSBs (Fig. 4g). To confirm this finding, we measured chromatin-bound RECQL4 in control or CUL4A-depleted U2OS cells after IR. IR enhances RECQL4 accumulation on chromatin in control cells, whereas knockdown of CUL4A greatly attenuated recruitment of RECQL4 (Fig. 4h). Taken together, the data demonstrate that the DDB1-CUL4A E3 ubiquitin ligase potentiates RECQL4's ability to participate in DSB repair.

**Phosphorylation enhances the helicase activity of RECQL4**. The enzymatic activity of RecQ helicases is tightly controlled by their interacting partners or by post-translational modifications in order to avoid promiscuous unwinding and annealing of DNA[47]. To explore whether CDK-dependent phosphorylation affects the

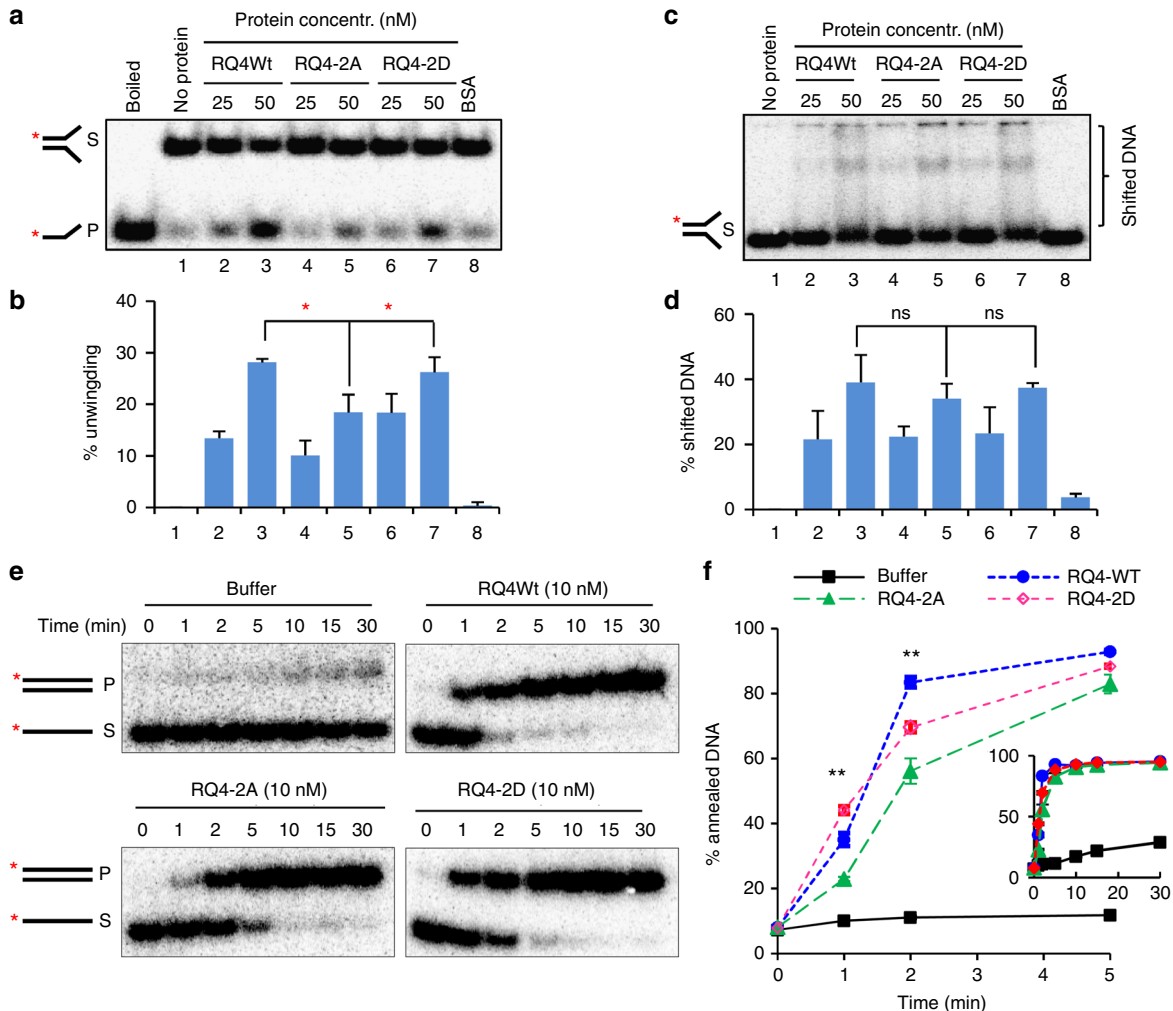

**Fig. 5** Phosphorylation on Ser89 and Ser251 stimulates helicase activity of RECQL4. **a**, **b** Helicase assay of RQ4Wt, RQ4-2A, and RQ4-2D on a Y-structured duplex substrate Mix3/4-19. 3×FLAG-tagged RECQL4 proteins were purified from HEK293T cell in presence of phosphatase and protease inhibitors. The purity of these proteins is shown in Supplementary Fig. 5a. RECQL4 (25 or 50 nM) proteins were incubated with 0.5 nM $^{32}$P-labeled mix3/4-19 for 30 min, and then the reactions were stopped. Representative image is shown in **a**, and the quantified data from three independent repeats are presented as mean ± s.e.m. with Student's $t$-test (**b**). **c**, **d** DNA-binding activity of RQ4Wt, RQ4-2A, and RQ4-2D. 0.5 nM Mix3/4-19 was incubated with 25 or 50 nM proteins for 10 min and DNA/protein complex was separated in native PAGE gel. Representative image is showed in **c**, and the quantified data from three repeats is shown in **d**, as mean ± s.e.m. with Student's $t$-test. **e**, **f** DNA annealing activity of RQ4Wt, RQ4-2A, and RQ4-2D. An 80-nucleotide long DNA substrate, G80, (0.5 nM) labeled with $^{32}$P on the 3′ end was mixed with complementary oligonucleotide C80 and 10 nM RECQL4 proteins. Samples of the reactions were withdrawn in a time-course manner, and terminated as described in Methods. Representative image is showed in **e**, and the quantified data presented as mean ± s.e.m. from four independent repeats is shown in **f**. $p$-values were determined by Student's $t$-tests. *$p<0.05$; **$p<0.01$. Insert shows full-time course of the experiment

biochemical activities of RECQL4, we purified 3×FLAG-tagged WT RECQL4, RQ4-2A, and RQ4-2D from HEK293T cells in the presence of phosphatase and protease inhibitors (Supplementary Fig. 5a), and tested for helicase and annealing activities of these proteins. RECQL4 processes 3′–5′ DNA unwinding activity[48,49], which is important for DNA end resection, HR, and for the prevention of senescence[28,34]. A Y-shaped duplex can be generated by MRN to facilitate recruitment of downstream factors and signaling[50]. RECQL4 is recruited by MRN to DSB before it promotes CtIP recruitment for the initial 5′ end resection[34]. Thus, a Y-shaped duplex seems an appropriate mimicking substrate to test DNA unwinding activity and DNA-binding activity of RECQL4 during end processing. Equimolar WT RECQL4 (50 nM) and phosphorylation-mimetic mutant RQ4-2D unwound significantly more Y-duplex substrate than RQ4-2A (Fig. 5a, b),

indicating stimulation of the helicase activity of RECQL4 by phosphorylation on Ser89/Ser251. RECQL4 has multiple DNA-binding domains and can bind to several DNA structures[51]. To measure the effect of Ser89/Ser251 phosphorylation on RECQL4's DNA-binding activity, we analyzed the DNA-binding activity of RECQL4 by gel mobility shift assay using the Y-shaped duplex and a duplex with 3′ ssDNA overhang. We observed that all three of the RECQL4 proteins shifted approximately 40% of the Y-shaped duplex (Fig. 5c, d). We also did not find any difference between the binding activities of the three RECQL4 proteins on the duplex with a 3′ overhang (Supplementary Fig. 5b). These data suggest that Ser89/Ser251 phosphorylation does not significantly change RECQL4's affinity for DNA. Uniquely among the RecQ helicases, RECQL4 possesses greater strand annealing than helicase activity[47,51]. Examining this activity, we found that

both WT RECQL4 and the phosphorylation-mimetic mutant RQ4-2D showed strong annealing activity at early time intervals, whereas RQ4-2A displayed a delayed rate of annealing as it shifted most of the substrate after 5 min (Fig. 5e, f). However, there were no differences between the proteins after 10 min. Thus, phosphorylation on RECQL4 Ser89/Ser251 is important for its rate of strand annealing. RECQL4 was previously shown to stimulate the nuclease activity of MRN complex in vitro[34]. Here we showed that RQ4-2D did not dramatically affect MRN-mediated cleavage of phiX174 ssDNA (Supplementary Fig. 5c), indicating that Ser89/Ser251 phosphorylation may not impact RECQL4's ability to stimulate MRN activity in vitro.

**Phosphorylation promotes RECQL4 in DNA end resection and HR.** CDKs regulate DNA end resection by phosphorylation of the major players in the process[4]. RECQL4 promotes 5′ end resection of broken DNA ends through the MRN/CtIP complex[34]. Here, we find that recruitment of RECQL4 is promoted by CDK1/2, which raises the possibility that this phosphorylation regulates the role of RECQL4 in DNA end resection. To assess this, DNA end resection was measured in AID-DIvA U2OS cells[52]. In line with our previous findings[34], depletion of endogenous RECQL4 reduced ssDNA content by 50% in AID-DivA U2OS cells, which is similar to the effect of depletion of MRE11 (Fig. 6a and Supplementary Fig. 6a). Expression of *RECQL4* siRNA-resistant WT

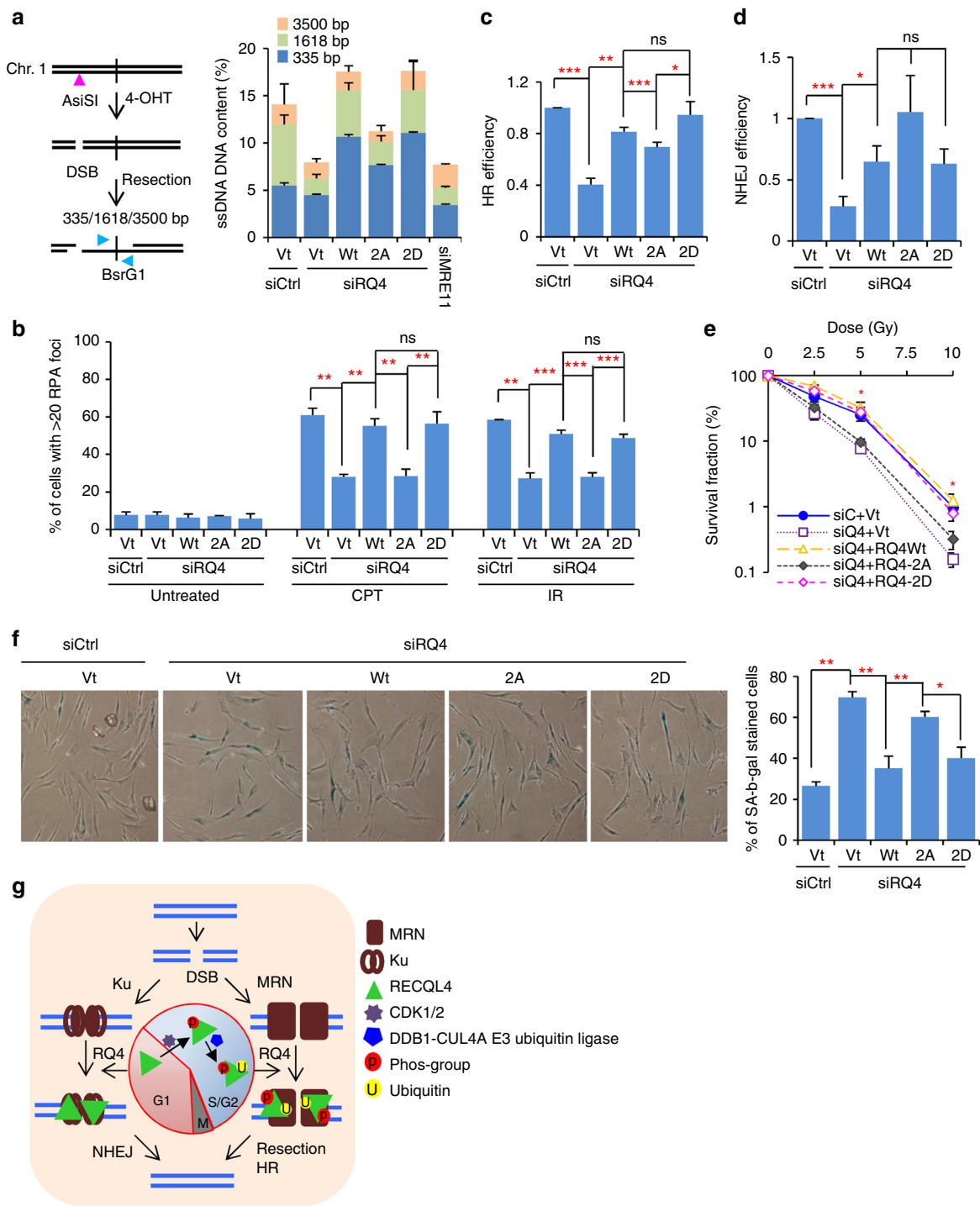

RECQL4 or RQ4-2D restored end resection, but RQ4-2A did not (Fig. 6a and Supplementary Fig. 6b). We also examined RPA focus formation after treatment with IR and the Topoisomerase I inhibitor Camptothecin (CPT) (Fig. 6b and Supplementary Fig. 6b, c). Without DNA damage treatment, the fraction of U2OS cells with >20 RPA foci was ~7% in cells. After treatment with CPT or IR, approximately 60% of control siRNA-treated cells had over 20 RPA foci per cell, while about 30% of the RECQL4-depleted cells did (Fig. 6b). This defect was corrected by expression of WT RECQL4 or RQ4-2D, but not by RQ4-2A (Fig. 6b). These findings together suggest that phosphorylation of RECQL4 by CDKs promotes DNA end resection.

DNA end resection is the initial step of HR for DSB repair[3]. Given the importance of RECQL4 Ser89/Ser251 phosphorylation for DNA resection, we next evaluated the role of CDK-mediated phosphorylation of RECQL4 in HR and NHEJ. HR efficiency was reduced by 60% after depletion of endogenous RECQL4 in DR-GFP U2OS cells, and restored by expression of WT RECQL4 or RQ4-2D (Fig. 6c and Supplementary Fig. 6d). Expression of RQ4-2A showed a partial rescue of HR in RECQL4-depleted cells, but significantly less than that of WT RECQL4 or RQ4-2D (Fig. 6c and Supplementary Fig. 6d). Depletion of RECQL4 also caused a significant loss of NHEJ, and expression of either WT RECQL4 or RQ4-2D rescued the loss of NHEJ to a similar extent (Fig. 6d and Supplementary Fig. 6e). Interestingly, RQ4-2A expression showed much more efficient NHEJ (Fig. 6d), which may be a consequence of increased interaction between Ku70 and RQ4-2A. These results indicate that CDK-mediated phosphorylation is important for RECQL4 in its pathway choice for DSB repair.

We next tested whether loss of DNA resection and HR resulted in reduced survival after IR treatment. Using a colony formation assay, we observed that control cells and RECQL4-depleted U2OS cells expressing WT RECQL4 or RQ4-2D showed similar survival curves after IR. In contrast, RECQL4-depleted U2OS cells complemented with vector or RQ4-2A expression showed significantly reduced survival (Fig. 6e and Supplementary Fig. 6f). These results indicate that phosphorylation of RECQL4 is important for cell survival after DNA damage.

RECQL4 preserves genome stability and prevents cellular senescence[28,29]. Since phosphorylation of RECQL4 is important for DNA repair and cell survival, we evaluated its role in senescence by measuring the senescence-associated β-galactosidase activity (SA-β-gal). In agreement with our previous findings[28], knockdown of RECQL4 increased the proportion of SA-β-gal stained cells from 26 to 70%, and expression of WT RECQL4 or RQ4-2D reduced the fraction of senescent cells to 35 and 40%, respectively (Fig. 6f and Supplementary Fig. 6g). In contrast, expression of RQ4-2A only reduced the percentage of SA-β-gal stained cells to 60% (Fig. 6f), indicating that phosphorylation of RECQL4 helps protect cells from senescence.

## Discussion

Many factors influence DSB repair pathways by specifically promoting or depressing one of the individual components. Interestingly, RECQL4 promotes both NHEJ and HR[33,34]. In this study, we investigated how RECQL4 promotes both pathways and coordinates the choice between NHEJ and HR. We report here that CDK1/2-mediated phosphorylation and DDB1-CUL4A-dependent ubiquitination of RECQL4 regulate DSB repair pathway choice in a cell cycle-dependent manner. RECQL4 is not targeted by CDKs in the G1 phase, where it interacts with Ku70 to stimulate NHEJ. However, during S/G2 phases, CDK1 and CDK2 both phosphorylate RECQL4 on Ser89/Ser251 and promote RECQL4's interaction with MRN/CtIP, which stimulates DNA resection and HR. In addition, we find that phosphorylation of RECQL4 at Ser89/Ser251 promotes its subsequent ubiquitination by the DDB1–CUL4A E3 ubiquitin ligase complex, facilitating RECQL4 participation in DSB repair. Thus, we report on a regulatory mechanism of RECQL4 in DSB repair pathway choice.

We show that the interactions of RECQL4 with the Ku heterodimer and the MRN/CtIP complex are cell cycle-regulated. RECQL4 preferably interacts with Ku70 in G1 phase and with MRN/CtIP in S/G2 phases. Given the essential roles of Ku and MRN/CtIP in NHEJ and HR[4], the cell cycle-dependent interaction of RECQL4 with Ku and MRN/CtIP coordinates the choice between NHEJ and HR. This coordination is controlled by CDK-dependent phosphorylation. CDK1 and CDK2 phosphorylate RECQL4 on Ser89/Ser251 during S/G2 phases, which is required for the MRE11/RECQL4 interaction. Ser89/Ser251 phosphorylation of RECQL4 stimulates MRE11-mediated DNA end resection and HR. Ku binds to DNA ends throughout the cell cycle and persistent binding of Ku to DNA ends attenuates DNA resection and HR[53]. Therefore, Ku is removed from DNA ends during S/G2 phases through multiple mechanisms, and this may partly explain the decreased interaction between RECQL4 and Ku in S/G2 cells. In G1 phase, phosphorylation of RECQL4 by CDKs is repressed, which results in low affinity of RECQL4 to MRN/CtIP. Meanwhile, DNA binding of Ku is not repressed[18,54,55]. Thus, RECQL4 primarily forms complexes with Ku to promote NHEJ. We demonstrate that the cell cycle status regulates the interaction between RECQL4 and MRN or RECQL4 and Ku through CDK1/2-dependent phosphorylation.

We identified a strong interaction between RECQL4 and DDB1, and detected DDB1-CUL4A-mediated ubiquitination of RECQL4. Remarkably, disruption of the DDB1-CUL4A E3 ubiquitin ligase by depleting either DDB1 or CUL4A greatly limits the recruitment of RECQL4 to DSBs (Fig. 4). Ubiquitination regulates protein–protein interactions and the local enrichment of DNA repair proteins at DSB sites[23]. DDB1-CUL4A-mediated ubiquitination may facilitate RECQL4's interaction with upstream proteins, promoting RECQL4 recruitment to DSBs. The Ddb1/

**Fig. 6** Ser89/Ser251 phosphorylation of RECQL4 promotes DNA end resection, HR, and prevents cellular senescence. **a** Analysis of DNA end resection in AID-DIvA U2OS cells. In AID-DIvA cells, the estrogen receptor-tagged endonuclease AsiSI translocates from cytoplasm to nucleus to induce DSBs in response to the presence of 4-Hydroxytamoxifen (4-OHT). The resected DNA is then quantified by TaqMan PCR. **b** RPA focus formation in U2OS cells. The cells were treated with 1 μM CPT for 1 h or 10 Gy IR with a 30-min recovery, and then processed for RPA32 immunostaining. The representative images are shown in Supplementary Fig. 6a. **c** HR assay in DR-GFP U2OS cells. **d** NHEJ assay in EJ5 U2OS cells. **e** Colony formation assay after IR stress in U2OS cells. **f** SA-β-gal staining assay in human primary fibroblasts GM5565, at passage 22. The images were taken under ×20 field. **g** Quantitative data from **f**. For all these assays, endogenous RECQL4 was depleted with *RECQL4* siRNA, and siRQ4-resistant plasmids expressing RQ4Wt, RQ4-2A, and RQ4-2D were transfected into RECQL4-depleted cells, and then submitted for resection assay, RPA foci, HR, colony formation and SA-β-gal staining assays. The data are presented as mean ± s.e.m. from at least three independent experiments. *p*-values were calculated with Student's *t*-test. ***$p < 0.001$; **$p < 0.01$; *$p < 0.05$. ns, not significant. **g** A model of how cell cycle-dependent phosphorylation regulates RECQL4 pathway choice and ubiquitination in DSB repair. RECQL4 plays important roles in both HR and NHEJ, which is tightly regulated across the cell cycle. RECQL4 is phosphorylated by CDK1 and CDK2 with their cyclin partners in S and G2 phases of the cell cycle, which regulates RECQL4's interaction with MRN to promote DNA end resection for HR. After CDK1/2-mediated phosphorylation, the DDB1-CUL4A E3 ubiquitin ligase ubiquitinates RECQL4, and further promotes recruitment of RECQL4 to DSBs. In G1 phase, RECQL4 interacts with the Ku complex to stimulate DSB repair by NHEJ

Cul4 E3 ubiquitin ligase complex monoubiquitinates H2B and promotes recruitment of ExoI for DNA resection in yeast[22]. Thus, it is probable that chromatin remodeling by DDB1-CUL4A-mediated ubiquitination affects downstream events in response to DNA damage, including RECQL4 recruitment to DSBs. Interestingly, phosphorylation on Ser89/Ser251 promotes ubiquitination of RECQL4 by enhancing the interaction between RECQL4 and DDB1. Since phosphorylation on Ser89/Ser251 is cell cycle-regulated, DDB1-CUL4A-mediated ubiquitination of RECQL4 likely also depends on the cell cycle. However, this needs further investigation. Moreover, inhibition of RECQL4 ubiquitination does not interfere with its CDK-mediated phosphorylation, suggesting the sequential modifications of RECQL4 by CDKs and then the DDB1-CUL4A E3 ubiquitin ligase through DSB repair. Sequential post-translational modifications have been shown to regulate other DNA metabolic proteins[56,57]. These post-translational modifications of RECQL4 illustrate the important role of this protein in repair of DSBs.

Here, we find that phosphorylation of RECQL4 on Ser89/Ser251 stimulates its DNA unwinding activity but not DNA-binding activity on a Y-shaped duplex (Fig. 5). MRN unwinds 15–20 pairs at the end of a Y-shaped DNA duplex and maintains the structure open, likely to facilitate recruitment of downstream enzymes to process the DNA broken ends[50]. Considering that MRN interacts with and recruits RECQL4, which depends on Ser89/Ser251 phosphorylation (Fig. 3), it is likely that RECQL4' helicase activity may contribute to maintaining the unwound single strands, facilitating recruitment of other DNA resection factors and end processing[34,58]. This may also explain why RECQL4 helicase activity is required to promote DNA end resection and HR[34].

Interestingly, the phosphorylation of RECQL4 also transiently increases strand annealing. The strand annealing property of RECQL4 is very prominent and may be of significant biological relevance. RECQL4 promotes recruitment of CtIP for the initial step of resection, but is not directly involved in the extensive resection step, since it dissociates quickly from DSBs[34]. When cells enter S phase, there is increased microhomology-mediated end joining[59], an alternative DSB repair pathway where strand annealing between resected ssDNA sharing microhomology is required. It is likely that RECQL4's strand annealing activity promotes microhomology-mediated end joining after the short resection by MRN/CtIP, and this is then facilitated by CDK1/2-mediated phosphorylation on Ser89/Ser251.

In this study, we demonstrate the importance of phosphorylation of RECQL4 on Ser89/Ser251 for DNA end resection and HR. This may reflect increasing recruitment of Ser89/Ser251-phosphorylated RECQL4 to DSBs via enhanced interaction between RECQL4 and MRE11. Given that RECQL4 promotes recruitment of CtIP to DSBs[34], enhanced recruitment of RECQL4 after CDK-dependent phosphorylation may recruit more CtIP, whose activity is also regulated by CDKs and is crucial to DNA resection[11,12]. Thus, the resection machinery efficiently operates with the coordination by CDK1/2.

Depletion of RECQL4 triggers senescence pathways[28]. Here, we show that CDK-dependent phosphorylation is important for RECQL4 to prevent cellular senescence in human fibroblasts. Cellular senescence has previously been associated with DNA damage[60]. Accordingly, we predict that defects in CDK1/2-mediated regulation of RECQL4 could lead to accumulation of unrepaired DNA damage, which triggers cellular senescence.

Based on the above data and on previously reported findings, we present a model for the role of RECQL4 in DSB repair pathway choice that is modulated by CDK1/2-mediated phosphorylation (Fig. 6g). If a DSB occurs in the G1 phase of the cell cycle, RECQL4 will primarily associate with the Ku complex to promote NHEJ. When cells move into the S/G2 phases of the cell cycle, CDK1/2 become active via the S/G2 phase cyclins, and phosphorylate RECQL4 on Ser89/Ser251, which then promotes DDB1-CUL4A-mediated ubiquitination of RECQL4. The sequential post-translational modifications of RECQL4 by CDK1/2 and the DDB1-CUL4A E3 ubiquitin ligase in S/G2 phase promotes recruitment of RECQL4 to DSBs, and stimulates DNA end resection by MRN/CtIP and HR.

## Methods

**Cell culture, knockdown DNA transfection, and γ radiation.** U2OS, HEK293T, and GM5565 cells were purchased from ATCC. DR-GFP U2OS[61] and EJ5 U2OS[36] were gifts from Dr. Xiaofan Wang (Duke University, USA) and Dr. Jeremy Stark (City of Hope, USA), respectively. AID-DIvA U2OS[52] is a gift from Dr. Gaëlle Legube (University of Toulouse, France) under Material Transfer Agreement between National Institute on Aging and Dr. Legube. All U2OS and HEK293T cells were cultured in DMEM medium with 10% fetal bovine serum (Sigma-Aldrich) and 1× penicillin/streptomycin (Gibco) in an atmosphere of 5% $CO_2$ at 37 °C. Primary human fibroblasts were cultured in MEM medium with 15% fetal bovine serum, 1× non-essential amino acid (Gibco), 1× vitamin (Gibco), 1× L-glutamine (Gibco), 1× penicillin/streptomycin. Lentivirus-mediated shRNA knockdown were achieved by incubating the lentivirus-expressing shRNA against DDB1 (Sigma-Aldrich, TRCN0000082856) or Control shRNA[28] with HEK293T cells for 2 days, and then selected for another 2 days with 2 µg mL$^{-1}$ puromycin (Invitrogen). For siRNA-mediated knockdown, siRNA was diluted with Opti-MEM medium (Gibco) to a final concentration at 30 nM, and then mixed with InterferIN (Polyplus transfection) according to the manufacturer's instructions. The cells were processed for the following experiments 4 days after transfection. The following siRNAs were used in this study, including siRQ4 (5′-CAAUACAGCUUACCGUACA-3′), siDDB1 (5′-UAACAUGAGAACUCUUGUC-3′), siCUL4A (5′-CCAUGUAA-GUAAACGCUUA-3′), and siMRE11 (5′-GCUAAUGACUCUGAUGAUA-3′). Transfection of plasmid DNA was conducted with JetPrime® (Polyplus), and ready for use after 24–36 h. γ rays were generated using a Cesium-137 source (Gammacell Exactor 40, Best Theratronics) and employed to irradiate cells at a dose of 10 Gy.

**Site-directed point mutation and plasmid construction.** Substitution of Ser89/Ser251 of RECQL4 with alanine or aspartic acid was carried out by PCR using pCMVtag4-RQ4-siR[28] as template, with the primers RQ4-S89A-PF: 5′-GGGCTGCGACCAAGGCTCCACAGCCTACGC-3′ and RQ4-S89-PR: 5′-GCGTAGGCTGTGGAGCCTTGGTCGCAGCCC-3′ for RQ4S89A; RQ4-S251A-PF: 5′-GCTGGGCTGGGGGGCCCCCACACGGATG-3′ and RQ4-S251-PR: 5′-CATCCGTGTGTGGGGGCCCCCCAGCCCAGC-3′ for RQ4S251A; RQ4-S89D-PF: 5′-GCGTAGGCTGTGGATCCTTGGTCGCAGCCC-3′ and RQ4-S89D-PR: 5′-GGGCTGCGACCAAGGATCCACAGCCTACGC-3′ for RQ4S89D; and RQ4-S251D-PF: 5′-GCTGGGCTGGGGGTCCCCCACACGGATG-3′ and RQ4-S251D-PR: 5′-CATCCGTGTGGGGGACCCCCAGCCCAGC-3′ for RQ4S251D. After confirmation by sequencing, the resulting plasmids expressing siRNA-resistant RECQL4-S89AS251A and RECQL4-S89DS251D were designated as pCMVtg4RQ4-2A-siR and pCMVtg4RQ4-2D-siR, respectively. For investigating the effect of phosphorylation events on Ser89 and Ser251 on recruitment to laser-induced DSB, *RECQL4* WT, *RQ4-2A*, and *RQ4-2D* were also cloned into pEGFP-C1 with restriction enzymes *Bgl*II and *Sal*I, and resulted in the plasmids pEGFP-C1-RQ4wt, pEGFP-C1-RQ4-2A, and pEGFP-C1-RQ4-2D.

**Synchronization and analysis of cell cycle.** U2OS cells were synchronized in G1 phase of cell cycle by DTB, and released into S or G2 phase as described[62]. Briefly, U2OS were incubated with 2 mM thymidine for 14 h, and released in fresh medium for another 12 h, then incubated with thymidine for another 12 h-thymidine block to block cell cycle at G1 phase. Enrichment of cells at S phase and G2 phase was achieved by releasing cells from DTB for 5 and 10 h, respectively. HEK293T cells were synchronized in the G1 phase by treating cells with 2 µg mL$^{-1}$ aphidicolin for 16 h as described[13]. For determining cell cycle, cells were washed and suspended with pre-chilled PBS, and then fixed with pre-chilled 70% ethanol. Fixed cells were stained with 0.1% Triton X-100, 10 µg mL$^{-1}$ propidium iodide, and 100 µg mL$^{-1}$ RNase in PBS, and then analyzed with an Accuri C6 flow cytometer. Flow J (version 10) software was used for analyzing the flow cytometry data.

**Western blotting and immunofluorescence microscopy.** For Western blotting, the proteins from IP or cell lysate were separated in 4–15% TGX gels (Bio-Rad), and then transferred to Polyvinylidene Difluoride (PVDF) membrane. The membranes were blocked with 5% non-fat milk or BSA, and then incubated with primary antibodies overnight at 4 °C. After washing with Tris buffered saline with 0.1% Tween-20, the HRP-conjugated secondary antibodies were then added to PVDF membrane. Finally, membranes were developed with Pierce ECL or Femto Western Blotting Substrate (Thermo-Fisher) and imaged with Bio-Rad ChemiDoc XRS imaging system. The intensity of bands was quantified with Image Lab 4.1 software (Bio-Rad). The following antibodies were used in Western blotting:

Anti-phospho-CDK substrate antibody (anti-p-SP antibody) (Cell Signaling, 9477), anti-RECQL4 antibody[28], anti-Mre11 antibody (Abcam, ab214), anti-DDB1 antibody (Abcam, ab109027), anti-CUL4A antibody (Abcam, ab72548), anti-Ubiquitin antibody (Abcam, ab134953), anti-CDK1 antibody (Abcam, ab133327), anti-CDK2 antibody (Abcam, ab32147), Anti-CyclinA2 (Abcam, ab38), Anti-Cyclin E antibody (Abcam, ab133266), anti-FLAG antibody (Sigma-Aldrich, F1864), anti-Tubulin antibody (Sigma-Aldrich, T5168), Anti-Ku70 antibody (Santa Cruz, sc-17789), anti-GFP antibody (Santa Cruz, sc-8334), Anti-phospho-H2AX (S139) antibody (EMD Millipore, 05-626), and Anti-CtIP antibody (Active Motif, 61141). All antibodies above were diluted 1:1000 for Western blotting, expect anti-DDB1 antibody (Abcam, ab109027), anti-Tubulin antibody (Sigma-Aldrich, T5168), and anti-FLAG antibody (Sigma-Aldrich, F1864), which were diluted 1:5000. The following secondary antibodies were used in this study with a dilution of 1:2000: Anti-mouse IgG Antibody (HRP-linked) and Anti-rabbit IgG Antibody (HRP-linked) from Cell Signaling, Rabbit TrueBlot®: Anti-Rabbit IgG HRP and Mouse TrueBlot® ULTRA: Anti-Mouse Ig HRP from Rockland Immunochemicals Inc. The uncropped blots were included as in Supplementary Fig. 7.

For immunostaining of endogenous RECQL4 at laser-induced DSBs, the U2OS cells were seeded to 3.5 mm Gridded glass bottom dishes (MatTek) 1 day before micro-point irradiation. DSBs were generated with a Stanford Research Systems (SRS) NL100 nitrogen MicroPoint system (Photonics Instruments) equipped to a Nikon Eclipse TE2000 spinning disk confocal microscope (Nikon Instruments Inc.) with Volocity software 6.3 (Perkin-Elmer). The cells in the marked areas were irradiated with 435 nm micro-point laser controlled by Volocity software. The treated cells were then fixed, permeabilized, and blocked with Fixation/Permeabilization Solution Kit (BD Biosciences). Endogenous RECQL4, γH2AX, and Cyclin A2 were immunostained with Anti-RECQL4 antibody (Santa Cruz, sc-16925) with a dilution of 1:200, Rabbit anti-γH2AX antibody (Abcam, Ab11174) with a dilution of 1:3000 and Mouse anti-CyclinA2 antibody (Abcam, ab16726) with a dilution of 1:3000. The secondary antibodies, Donkey anti-Goat IgG (H+L) Alexa Fluor 488 (Invitrogen, A-11055); Donkey anti-Mouse IgG (H+L) Alexa Fluor 594 (Invitrogen, A-21203), and Donkey anti-Rabbit IgG (H+L) Alex Fluor 647 (Invitrogen, A-31573) were used with a dilution of 1:1000. After staining, the cells were mounted in Prolong Gold Antifade Mounting Solution with DAPI (Invitrogen). The images were captured with Nikon TE2000 confocal microscope and analyzed with Volocity software v6.3.

For measuring RPA foci, U2OS cells were transfected with siRQ4 to deplete endogenous RECQL4, and 2 days later transfected with the plasmids expressing WT RECQL4, RQ4-2A, and RQ4-2D, as well as vector. After 2 days, the cells were with a 10 Gy γ radiation, and then were allowed to recover for 30 min before processed for immunostaining. For CPT treatment, the cells were incubated with 1 μM CPT for 1 h before processed for immunostaining. For immunostaining, the cells were fixed, permeabilized and blocked with Fixation/Permeabilization Solution Kit (BD Biosciences). RPA was stained with Anti-RPA32 antibody (EMD Millipore, NA19) with a dilution of 1:200, then stained with Goat anti-mouse IgG (H+L) Alex Fluor 488 (Invitrogen, A32723), and mounted with Prolong Gold Antifade Mounting Solution with DAPI. RPA foci were observed with Zeiss Axiovert 200 M fluorescence microscope equipped with AxioCam HRM camera. Over 200 M for each sample were calculated, and the results were presented mean ± s.e.m. from three independent experiments.

**IP and mass spectrometry**. For detecting phosphorylation and ubiquitination of RECQL4, cells were lysed in a denatured condition and then processed for IP as described with some modifications[63]. For phosphorylation detection, the plasmids pCMVtag4-RQ4-siR and pCMVtag4-RQ4-2A-siR expressing 3×FLAG-tagged WT RECQL4 and RQ4-2A were transfected into HEK293T cells. The cells were lysed in lysis buffer 1 (20 mM Tris 8.0, 150 mM NaCl, 2% SDS, 1× phosphatase inhibitors cocktails 2 (Sigma-Aldrich), 1× protease inhibitor cocktails (Thermo Fisher)), and heated at 95 °C for 10 min, followed by sonication. The lysate was then diluted 10 times with dilution buffer (20 mM Tris 8.0, 150 mM, 1% NP-40, 2 mM EDTA, 1× phosphatase inhibitors cocktails 2, 1× protease inhibitor cocktails), and incubated on a rotor at 4 °C for 1 h. After removing cell debris by centrifuging, the lysates were incubated with magnetic M2 FLAG beads (Sigma-Aldrich) overnight at 4 °C. Beads were washed three times with Washing buffer 1 (20 mM Tris 8.0, 1 M NaCl, 1 mM EDTA, 1% NP-40) and then mixed with 2×SDS sample buffer and processed for SDS-PAGE, followed by Western blotting or mass spectrometry. For mass spectrometry, the purified FLAG-RQ4 was isolated by SDS-PAGE, and stained with SimplyBlue™ SafeStain (Invitrogen). The band with FLAG-RQ4 was sliced and submitted to Harvard Taplin Mass Spectrometry Facility.

Ubiquitination of RECQL4 was investigated as previously described[63]. Briefly, DDB1-depleted HEK293T cells and control cells were transfected with plasmids expressing 3×FLAG-tagged RECQL4 and HA-tagged ubiquitin. Cell lysate was prepared as described above, and 1 mg protein from each sample was incubated with Roche anti-HA affinity matrix to pull down ubiquitinated proteins. Ubiquitinated RECQL4 was detected with anti-FLAG antibody in Western blotting. For detecting ubiquitination of endogenous RECQL4 in U2OS cells, all procedures were same as above, except that RECQL4 was captured with anti-RECQL4 antibody and Protein A-Agarose beads (Santa Cruz). 10 μM RO-3306 (SelleckChem) and 10 μM CDK2i-III (EMD Millipore) or DMSO (Sigma-Aldrich) were incubated with U2OS cells when applied.

To investigate the RECQL4-interacting proteins, HEK293T cells expressing GFP or GFP-tagged RECQL4 were suspended in lysis buffer 2 containing 40 mM Tris-HCl pH 7.4, 100 mM NaCl, 2 mM MgCl₂, 0.2% NP-40, 0.3% Triton 100, 1× protease inhibitor cocktail, 1× phosphatase inhibitor cocktail 2, 20 U mL⁻¹ Benzonase (Novagen). To exclude DNA/RNA-bridged associations, all IPs for protein–protein interaction were performed in the presence of Benzonase. Then the cells were sonicated on ice. After removing the cell debris by centrifuging, the supernatant containing 2 mg of protein were incubated with 20 μL GFP-TRAP beads (ChromoTek) overnight at 4 °C. The beads were washed with washing buffer 2 (20 mM Tris 7.4, 150 mM NaCl, 0.2% Triton X-100) for five times and subjected for mass spectrometry or Western blotting. For mass spectrometry, the proteins on the beads were eluted and precipitated with EMD Millipore ProteoExtract® Protein Precipitation Kit, and then submitted to Harvard Taplin Mass Spectrometry Facility for mass spectrometry analysis. The details of mass spectrometry and data search are listed in the Supplementary Data 1. For inhibitor treatments, cells were pretreated with DMSO, 20 μM CDK2i-2 (Santa Cruz), or 20 μM ATM inhibitor KU55933 (SelleckChem) for 4 h, and then treated with IR for 10 Gy. Next, cells were recovered in normal cell culture conditions for 5 min before being processed for IP as described above. For the IP investigating interaction between endogenous RECQL4 and endogenous MRE11, U2OS cell lysate were prepared using Lysis buffer 2, and anti-MRE11 antibody or normal mouse IgG were added to lysate. The protein A/G agarose beads were used to capture the MRE11 antibody and mouse IgG. After washing with washing buffer 2, IP product was submitted for WB analysis.

To map the interaction region of RECQL4 with CDK1 and CDK2, 3×FLAG-tagged RECQL4, RQ4NT4(1-427AA) and RQ4RQ/CT(427-1208AA) were expressed in HEK293T cells by transfecting the plasmids pCMVtag4A-RQ4-siR, pCMVtag4A-RQ4-1-427-NLS-3×FLAG, and pCMVtag4A-RQ4-427-1208-NLS-3×FLAG[34], as well as control vector into HEK293T cells, respectively. Twenty-four hours later, the cells were washed with cold PBS, harvested, suspended in lysis buffer 2 and then sonicated on ice. After removing the cell debris by centrifuging, the supernatant containing 2 mg of protein were incubated with 20 μL M2 FLAG beads overnight at 4 °C. The beads were washed with washing buffer 2 as described above and subjected for Western blotting.

**Protein purification**. For purification of 3×FLAG-tagged RQ4Wt, RQ4-2A, and RQ4-2D from HEK293T cells, the plasmids pCMVtag4-RQ4-siR, pCMVtg4RQ4-2A-siR, and pCMVtg4RQ4-2D-siR were transfected into HEK293T cells, and purification was performed as described[34]. Briefly, cells were suspended in lysis buffer 3 (50 mM HEPES 7.4, 500 mM NaCl, 0.5% NP-40, 10% Glycerol, 1× phosphatase inhibitor cocktail, 1× protease inhibitor cocktail), and sonicated on ice. 3×FLAG-tagged proteins were captured by M2-FLAG beads, which were washed with washing buffer 3 (50 mM HEPES pH 7.4, 1 M NaCl, 0.5% NP-40, 10% Glycerol, 1× phosphatase inhibitor cocktail, 1× protease inhibitor cocktail), lysis buffer 3, and washing buffer 4 (50 mM HEPES 7.4, 150 mM NaCl, 0.5% NP-40, 10% Glycerol), washing buffer 5 (50 mM HEPES 7.4, 150 mM NaCl, 10% Glycerol), and washing buffer 6 (50 mM HEPES 7.4, 80 mM KCl, 10% Glycerol). Finally, proteins were eluted with washing buffer 6 containing 250 μg mL⁻¹ 3×FLAG peptides. Protein purity was examined by Coomassie blue gel staining after SDS-PAGE and the concentration was measured with Bio-Rad Protein Assay Kit 1.

**In vitro phosphorylation**. 3×FLAG-tagged WT RECQL4 and RQ4-2A were expressed in HEK293T cells, and purified as decribed above the beads were treated with λPP (New England Biosciences) at 30 °C for 30 min. The beads were further sequentially washed with washing buffer 3, lysis buffer 3, washing buffer 5, and 1× in vitro phosphorylation buffer containing 20 mM HEPES pH 7.4, 5 mM MgCl₂, 5 mM MnCl₂, 2 mM ATP, 1 mM DTT, and 1 mM sodium pyruvate. 100 ng CDK1/CyclinA (Thermo Fisher), CDK2/CyclinA or CDK2/CyclinE (EMD Millpore) was incubated with the beads containing λPP-treated 3×FLAG-RECQL4 or RQ4-2A in in vitro phosphorylation buffer. Reactions were conducted at 30 °C for 30 min, and stopped by adding 2× SDS sample buffer and processed for Western blotting.

**Laser-induced DNA damage and real-time recruitment of fluorescent proteins**. Generation of laser-induced DSB and recruitment of fluorescence protein-tagged proteins were performed with a SRS NL100 nitrogen MicroPoint system equipped to a Nikon Eclipse TE2000 spinning disk confocal microscope with Volocity software 6.3 (Perkin-Elmer)[34,64]. For treatment of CDK inhibitors, U2OS cells expressing GFP/YFP-tagged RECQL4 and YFP-MRE11 were pre-incubated with DMSO, 20 μM Roscovitine (Santa Cruz), 10 μM CDK1 inhibitor RO3306, 10 μM CDK2 inhibitor 2-III (EMD Millipore) for 4 h under standard cell culture conditions, and then subjected to micro point laser-irradiation. To study the function of CDK-dependent phosphorylation of Ser89 and Ser251 on RECQL4 recruitment to DSBs, GFP-tagged WT RECQL4, RQ4-2A, and RQ4-2D were expressed by transfecting the plasmids pEGFP-C1-RQ4Wt, pEGFP-C1-RQ4-2A, and pEGFP-C1-RQ4-2D into U2OS cells, respectively. For CDK 1and CDK2 inhibition, the U2OS cells were incubated with RO3306 and CDK2 inhibitor-III for 4 h before laser irradiation. The recruitment of these proteins was conducted as described above. The fluorescence intensity of the damaged area was measured with Volocity imaging software and normalized to that of a control area. The

results are presented as mean ± s.e.m., and P -values were measured with Student's t-test.

**Biochemical assays**. For helicase and DNA binding assays, a Y-shaped duplex DNA substrate Mix 3/4-19 was constructed by annealing γ-$^{32}$P-labeled Mix3-19 at 5′ end (5′-TTTTTTTTTTTTTTTTTGAGTGTGGTGTACATGCAC-3′) with unlabeled oligo Mix4-19 (5′-GTGCATGTACACCACACTCTTTTTTTTTTTTTT-3′). For helicase assay, 25 or 50 nM RQ4-Wt, RQ4-2A, or RQ4-2D was incubated with 0.5 nM Mix3/4-19 in the helicase reaction buffer (25 mM HEPES, 7.4, 5 mM MnCl$_2$, 5 mM ATP, 1 mM DTT, 1 nM cold Mix 319, 100 ng mL$^{-1}$ BSA and 5% Glycerol) for 30 min at 37 °C. The reactions were terminated by adding 3× helicase stopping buffer (0.9% SDS, 50 mM EDTA, 0.01% bromophenol blue). For electrophoretic mobility shift assays, 0.5 nM Mix3/4-19 was mixed with 25 or 50 nM RQ4-Wt, RQ4-2A, or RQ4-2D in the helicase reaction buffer without 5 mM ATP and 1 nM cold Mix 3-19, and incubated at 37 °C for 10 min. DNA/protein mixture was then separated on 10% native PAGE (Acrylamide/Bis-Acrylamide: 19:1 for helicase assay, 29:1 for electrophoretic mobility shift assay). For making the duplex with 3′ ssDNA overhang, two following oligos were used, 5′-ACTTGAATGCGGCTTAGTATGCATTGTAAAACGACGGCCAGTGC-3′ and 5′-AATGCATACTAAGCCGCATTCAAGT-3′. The DNA strand annealing activity of WT and mutant RECQL4 were measured using single-stranded complementary synthetic oligonucleotides (0.5 nM) of 80-mers G80 (5′-GCTGATCAACCCTACA TGTGTAGGTAACCCTAACCCTAACCCTAAGGACAACCCTAGTGAAGCTT GTAACCCTAGGAGCT-3′) and C80 (5′-AGCTCCTAGGGTTACAAGCTTCAC TAGGGTTGTCCTTAGGGTTAGGGTTAGGGTTACCTACACATGTAGGGT TGATCAGC-3′)[65] with G80 labeled on the 5′ end using [γ-$^{32}$P] ATP using T4 polynucleotide kinase. Annealing reactions (10 μL) were carried out in a time-dependent manner (0, 1, 2, 5, 10, 15, and 30 min) at 37 °C in buffer (30 mM Tris-HCl pH 7.5, 50 mM KCl, 1 mM DTT, 5 mM MgCl$_2$, BSA 100 μg ml$^{-1}$), the reaction was started with the addition of G80 and stopped by addition of stopping buffer (30 mM EDTA, 0.9% SDS, 30 mM Tris-Cl pH 8.0, 30% glycerol, 0.05% bromophenol blue, and 0.05% xylene cyanol). The reaction products were separated by electrophoresis on a 10% native polyacrylamide gel. Images were captured on a GE Typhoon phosphorimager, and quantified with ImageQuantTL.

For nuclease assay, 20 nM MRN or nuclease-dead MRN-ND were pre-incubated with 20 nM RQ4-2D or BSA (New England Biolabs) on ice for 5 min, and then the reactions were initiated by adding the reaction buffer (20 mM MOPS, pH 7.2, 1 mM DTT, 5 mM MgCl$_2$, 5 mM MnCl$_2$, 1 mM ATP, and 50 ng PhiX174 circular ssDNA; New England Biolabs). After a 3-h incubation at 37 °C, the reactions were stopped with 1/10 (v/v) stopping buffer (2% SDS, 100 mM EDTA, 1 mg mL$^{-1}$ proteinase K). The DNA separated on a 0.8% 1× TAE agarose gel, visualized by SYBR Gold (Life Technologies) with Chemidoc™ XRS+ system and quantified with Bio-Rad Image Lab™.

**DNA resection assay, HR assay, and NHEJ assay**. For in vivo 5′ end resection, endogenous RECQL4 was depleted by *RECQL4* siRNA in the AID-DIvA U2OS cells, and the plasmids expressing 3×FLAG-tagged WT RECQL4, RQ4-2A, and RQ4-2D as well as vector were transfected to the RECQL4-depleted cells. The cells were treated with 4-Hydroxytestosterone (4-OHT) for 4 h for cellular translocation of *Asi*SI endonuclease from cytoplasm to nucleus to make DSBs. Genomic DNA was prepared using Blood & Cell Culture DNA Mini Kit (Qiagen). After treatment of RNase H (New England Biosciences), 3 μg genomic DNA was mock digested or digested with *Bsr*GI or *Hin*dIII. Then, ssDNA generated by the 5′ resection was examined by qPCR with TaqMan probes[66] with 18 ng of digested DNA as templates, then quantified as ssDNA per cent, by calculating the change in Ct values of mock and restriction enzyme digested DNA using the following equation: ssDNA % = $1/(2^{(\Delta Ct-1)} + 0.5) \times 100$. The data were presented with mean ± s.e.m. from three independent experiments.

The HR repair and NHEJ assays were performed in DR-GFP U2OS cells and EJ5 U2OS cells, respectively. Three days after siRNA transfection for knocking down endogenous RECQL4, 0.5 μg pCMVtag4-RQ4-siR Wt, 2A, 2D or vector, as well as the plasmids expressing I-SceI endonuclease and DsRed were transfected into 1 × 10$^6$ DR-GFP U2OS cells or EJ5 U2OS cells with Lonza Solution V with program X-001. Four days after the transfection, the cells were submitted for flow cytometry with Accuri C6 Flow Cytometer. The results are presented as mean ± s.e.m. from four independent experiments with p-values determined by Student's t-test.

**Colony formation assay**. The endogenous RECQL4 was depleted in U2OS cells by siRNA-mediated knockdown as mentioned above. Two days after siRNA transfection, the cells were then transfected with the plasmids expressing WT RECQL4, RQ4-2A, RQ4-2D or vector. Two days later, the cells were seeded into 6 cm dishes and irradiated after 12 h. The cells were allowed to recover for 10 days, and then fixed and stained with 2% methylene blue in 5% ethanol at room temperature for 15 min. The dishes were gently washed with water, and dried in air. The colonies with over 50 cells were counted. The results are presented as mean ± s.e.m. from three independent experiments.

**SA-β-gal staining**. Normal human fibroblast cells GM5565 at passage 22 were transfected with RECQL4 siRNA first, and 24 h later siRQ4-resistant plasmids pCMVtag4-RQ4-siR Wt, 2A, 2D or vector were transfected into siRQ4-transfected GM5565 cells, and the vector was transfected to siCtrl-treated GM5565 cells. After another 3 days, cells were submitted for SA-β-gal staining using Senescence Cells Histochemical Staining Kit (Sigma-Aldrich), per manufacturer's instructions.

**Data availability**. The mass spectrometry data for identifying phosphorylation sites are included in Fig. 2c and Supplementary Fig. 1. The mass spectrometry data for identifying RECQL4-interacting proteins are listed as Supplementary Data 1. All uncropped Western blots are included in Supplementary Fig. 7. The other data that support the findings of this study are available from the corresponding author upon request. A Life Sciences Reporting Summary for this paper is available.

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

## Acknowledgements

We thank Dr. Xiaofan Wang, Dr. Jeremy Stark, Dr. Gaëlle Legube, and Dr. Petr Cejka for DR-GFP U2OS, EJ5 U2OS, AID-DIVA U2OS, and MRN proteins. We also thank Alfred May and Dr. Janapriya Saha for IR operation, and Drs. Jong-Hyuk Lee and Tyler Demarest for their critical comments. This research was supported mainly by the Intramural Research Program of the NIH, National Institute on Aging (V.A.B.), and also by grants from the National Institutes of Health CA092584 and CA162804 (A.J.D.).

## Author contributions

H.L., R.A.S., and V.A.B. developed concept and designed experiments. H.L. performed the majority of the experiments with help from J.K.D., M.O., P.P.H., T.K. and J.T., R.A.S. and H.L. performed the assay for measuring recruitment of RECQL4 to micro-point laser-induced DSBs; P.K. performed strand annealing assay; H.L., R.A.S., D.L.C., A.J.D., and V.A.B. prepared the manuscript.

## Additional information

**Competing interests:** The authors declare no competing financial interests.

