## [Peer Review File · Nature Communications]

Reviewers' Comments:

Reviewer #1:

Remarks to the Author:

In this manuscript, Lu et al. demonstrated that while helicase RECQL4 promotes both NHEJ and HR-mediated DSB repair pathways, the DSB pathway choice is regulated by cell cycle. The authors substantiate their claim by identifying two serine residues in the N-terminus of RECQL4 to be phosphorylated by CDK1/2 during S/G2 phase. The phosphorylation of these two sites promotes MRE11-RECQL4 interaction and recruitment to DSB. The authors also demonstrated that RECQL4 is also ubiquitylated and that the ubiquitylation also required to promote RECQL4 localization to laser-induced damage. The authors further demonstrated that these PTMs on RECQL4, regulated by cell cycle, promotes helicase activity that enhances resection and promotes HR pathway.

While the authors' findings presented important regulatory mechanisms of a helicase that determines DSB repair pathway choice, there are several major points that authors need to address to render their conclusions convincing.

Figure 1: The authors' data are very clear that RECQL4 is required for both NHEJ and HR efficiency. In Figure 1F, the authors claimed that interaction of RECQL4 with resection machinery protein MRE11 is enriched while interaction with Ku70 is diminished in S/G2 from IP of GFP-RECQL4. Thus, the enriched interaction with MRE11 in S/G2 cells implicate RECQL4 could promote end resection, thereby HR in that specific phase of the cell cycle. I am wondering if authors could also substantiate their claim by also showing whether there is enriched interaction with other components of end resection machinery, including RAD50, NBS1, CTIP, etc.

Figure 2: The authors identified two serine sites in RECQL4 that are putatively CDK phosphorylated, showed that this phosphorylation is enriched S/G2 (Fig 2G). Although the authors have shown flow cytometry profile of the cells released from double thymidine to demonstrate that phosphorylation is enriched in S and G2, the flow cytometry quantification depends on the gating criteria which can be subjective. Hence it would be more convincing if cell cycle progression could also be shown through the diminishing level of cyclin E as cells progress through S phase; and rising level of cyclin A as cells progress through G2 phase.

Figure 3: The authors aim to demonstrate that the CDKs interacts with RECQL4 and that CDK1/2 are the kinases that phosphorylate RECQL4. In figure 3F and 3G, the authors showed that expression of phosphomimetic mutant of RECQL4 phenocopies wildtype and phosphonull mutant phenocopies CDKi inhibition in terms of GFP-RECQL4 localization to laser-induced DNA damage. However, to demonstrate that it is the CDK-mediated phosphorylation of RECQL4 that promotes its recruitment to DNA damage, the authors should demonstrate that phosphomimetic mutant RECQL4-2D can rescue the localization defect rendered by the inhibition of CDKs. Furthermore, the authors demonstrate in Fig 3H that specific inhibition of CDK diminished the interaction of RECQL4 and MRE11, thus suggesting CDK-mediated phosphorylation of RECQL4 is required for its interaction with MRE11, the authors should include results from IP of RECQL4 phosphonull and phosphomimetic mutant to verify that direct phosphorylation of RECQL4 is required for its interaction with MRE11.

Figure 4: Here authors intend to demonstrate that the E3 ubiquitin ligase complex that found to interact with RECQL4, shown by IP-MS, also promotes the ubiquitylation of RECQL4 and that this ubiquitylation is dependent on phosphorylation. Furthermore, the ubiquitylation of RECQL4 is also required for its localization to damaged chromatin. The authors demonstrated the requirement of phosphorylation for ubiquitylation through either siRNA-mediated depletion of E3 ubiquitylating enzyme complex, or expression of phosphonull form of RECQL4 that resulted in diminishing level of ubiquitylation by immunoblotting, compared to expression of wildtype. However, for a more rigorous and specific proof, the authors should purify WT and phosphonull mutant and detect ubiquitylation sites, ie putative lysine residues, by MS/MS, and subsequently express

ubiquitylating-null mutant to look for resulting defect in RECQL4 localization in response to laser induced damage.

Figure 5: The authors showed that phosphorylation of RECQL4 stimulates its enzymatic activities on DNA, including helicase and annealing activities. The authors demonstrated the helicase activities and RECQL4's ability to bind DNA using splayed-arm/Y-structure duplex DNA oligo structure. I think the authors could better substantiate their conclusions regarding RECQL4 stimulating end resection with MRE11 during S/G2 by demonstrating that RECQL4 display substrate binding and helicase activity preference specifically towards duplex with only 3' ssDNA overhang, which better mimics physiological substrate for end resection; as oppose to Y structure, which supports RECQL4 function in the context of replication. Testing alternative substrate to identify preferred substrate of RECQL4 could also render more obvious DNA binding product that that shown in Fig 5C.

Figure 6: Here the authors demonstrates phosphorylation of REQL4 is required for end resection, thereby HR efficiency. Although the results for WT and phosphomimetic mutant are clear compare to phoshonull mutant (Fig 6 A, B, C, D), it would be important for authors to include MRE11 depletion as a control for these experiments to compare the level of end resection defect manifested by phosponull mutant since one of authors key points is that phosphorylation of RECQL4 is required for MRE11 interaction hence phosponull RECQL4 should phenocopy MRE11 depletion.

Overall, the authors showcased results that demonstrated important regulatory mechanisms, through post-translational modification of RECQL4 helicase, that promotes HR during S/G2. If the authors could include results from suggested experiments mentioned herein, the conclusion would be better supported.

Reviewer #2:

Remarks to the Author:

In their manuscript entitled 'Cell cycle-dependent phosphorylation and ubiquitination regulates RECQL4 pathway choice in DNA double-strand break repair', Lu et al. describe the role of RECQL4 phosphorylation and ubiquitylation in cell cycle regulation of double strand break repair. The authors claim that CDK-induced phosphorylation of RECQL4, which is followed by ubiquitination by CUL4-complexes, favors binding to MRE11 hence stimulating DNA end resection in S-phase, while in G1 phase, RECQL4 binds to the KU complex and stimulates NHEJ.

The authors nicely show RECQL4 is phosphorylated by CDK1/2 during S-phase, thereby increasing its binding to MRE11 and recruitment to DSBs. However, they previously already showed that RECQL4 stimulates Mre11-dependent end-resection and homologous recombination, severely limiting the novelty of the current work. With regard to this latter point, the following major points should be addressed to warrant publication of the manuscript:

1. whether this phosphorylation affects KU binding (and thereby explains the shift in complex formation from G1 to S phase) and NHEJ repair
2. mechanistic insight into how CDK-mediated phosphorylation drives Mre11-dependent end-resection and DSB repair by homologous recombination
3. what the effect of the CUL4-mediated ubiquitylation is on DSB repair pathway choice

The paper should be improved by adding the following experiments to test:

- RECQL4-2D mutants for KU binding and DSB recruitment in G1
- if expression of RECQL4-2D mutant affects NHEJ
- whether the RECQL4-2A mutant shows reduced interaction with Mre11
- whether endogenous RECQL4 and Mre11 interact in reciprocal IPs and whether inhibition of CDK1/2 affects this interaction, since all experiments were done following overexpression of FLAG-

tagged RECQL4

- whether inhibition of CDK1/2 also affect phosphorylation and damage recruitment of endogenous RECQL4
- whether Mre11 is still recruited following CDK1/2-inhibition
- if RECQL4-2D stimulates MRE11 activity in vitro
- if endogenous RECQL4 is ubiquitylated by DDB1
- the effect of RECQL4 ubiquitylation on MRE11 and KU binding
- the effect of RECQL4 ubiquitylation on RECQL4 catalytic activity (experiments of Fig. 5 upon purification of RECQL4 from DDB1 siRNA treated cells, or better test a RECQL4 mutant that cannot be ubiquitylated)
- if the RECQL4-2A mutant shows a reduced DDB1 interaction, to show how RECQL4 phosphorylation could lead to DDB1-dependent ubiquitylation

Minor points:

- At page 6 of the manuscript, first paragraph, the authors write 'which is consistent with previous reports' (the DR-GFP data). Since they are self-citing their own paper, which shows the same results, using the same assay, I don't find it a strong argument for consistency.
- Figure 1 seems to contain data that have already been reported before by the authors.
- Given the role of RECQL4 on DNA replication, does RECQL4 knock-down or knock-out affect cell cycle, thereby affecting HR rate?
- Figure 1A-B, the effect of RECQL4 knockdown on HR and NHEJ is comparable, while the knockdown is not (poor knockdown in A, highly efficient knockdown in B). How can this be explained?
- It is unclear to me whether the RECQL4 recruitment to laser tracks experiments in figure 3f/g and 4g are based on S-phase cells only. If not, I wonder whether the data is skewed by cells in G1 binding to DSBs via KU.
- Figure 4e, confirm the effect of RECQL4 phosphorylation on ubiquitylation by testing ubiquitylation of RECQL4 after CDK1/2 inhibition.
- In the discussion, the authors suggest that DDB1/CUL4-mediated ubiquitylation of RECQL4 depends on the cell cycle since it depends on the cell cycle regulated phosphorylation. It would be of interest to confirm this hypothesis.

Response to the Reviewers' comments:

Reviewer #1 (Remarks to the Author):

In this manuscript, Lu et al. demonstrated that while helicase RECQL4 promotes both NHEJ and HR-mediated DSB repair pathways, the DSB pathway choice is regulated by cell cycle. The authors substantiate their claim by identifying two serine residues in the N-terminus of RECQL4 to be phosphorylated by CDK1/2 during S/G2 phase. The phosphorylation of these two sites promotes MRE11-RECQL4 interaction and recruitment to DSB. The authors also demonstrated that RECQL4 is also ubiquitylated and that the ubiquitylation also required to promote RECQL4 localization to laser-induced damage. The authors further demonstrated that these PTMs on RECQL4, regulated by cell cycle, promotes helicase activity that enhances resection and promotes HR pathway. While the authors' findings presented important regulatory mechanisms of a helicase that determines DSB repair pathway choice, there are several major points that authors need to address to render their conclusions convincing.

Response: We thank the reviewer for these comments. In response to these and the comments from Reviewer 2, we have performed several new experiments which are described in detail below.

Figure 1: The authors' data are very clear that RECQL4 is required for both NHEJ and HR efficiency. In Figure 1F, the authors claimed that interaction of RECQL4 with resection machinery protein MRE11 is enriched while interaction with Ku70 is diminished in S/G2 from IP of GFP-RECQL4. Thus, the enriched interaction with MRE11 in S/G2 cells implicate RECQL4 could promote end resection, thereby HR in that specific phase of the cell cycle. I am wondering if authors could also substantiate their claim by also showing whether there is enriched interaction with other components of end resection machinery, including RAD50, NBS1, CtIP etc.

Response: MRE11-RAD50-NBS1 (MRN) initiates DNA end resection with CtIP for homologous recombination during S/G2 phases. We had previously reported that RECQL4 promotes complex formation of MRN/CtIP (Lu et al., Cell Reports, 2016, 16:161-173). In the revised manuscript Fig 1f, as suggested by the reviewer, we have now performed a new immunoprecipitation (IP) experiment to show that there is enriched interaction of RECQL4 with CtIP, RAD50 and NBS1 in S/G2 cells after IR stress, in addition to that with MRE11. Consistently, the interaction between RECQL4 and Ku70 is enhanced in G1 cells, but repressed in S/G2 cells. This data strongly supports the cell cycle-dependent pathway choice of RECQL4 for DSB repair. Please see the new Fig. 1f and rewritten paragraph in Page 7.

Figure 2: The authors identified two serine sites in RECQL4 that are putatively CDK phosphorylated, showed that this phosphorylation is enriched S/G2 (Fig 2G). Although the authors have shown flow cytometry profile of the cells released from double thymidine to demonstrate that phosphorylation is enriched in S and G2, the flow cytometry quantification depends on the gating criteria which can be subjective. Hence it would be more convincing if cell cycle progression could also be shown through the diminishing level of cyclin E as cells progress through S phase; and rising level of cyclin A as cells progress through G2 phase.

Response: We have now performed the Western blot analyzing the level of Cyclin A and Cyclin E in cell cycle-synchronized 293T and U2OS cells. We do find high levels of Cyclin E and low levels of Cyclin A in the cells immediately after double thymidine block or aphidicolin. After release for 5 or 10 hrs from thymidine or aphidicolin block, the level of Cyclin A increases, while the Cyclin E level decreases. These Western blotting data are in the line with the results from flow cytometry, together suggesting that the cell-cycle synchronization in both U2OS cells and HEK293T cells is consistent and stable. Please see the new data in Supplemental fig. 2b and 2d and related text in Page 9.

Figure 3: The authors aim to demonstrate that the CDKs interacts with RECQL4 and that CDK1/2 are the kinases that phosphorylate RECQL4. In figure 3F and 3G, the authors showed that expression of phosphomimetic mutant of REQL4 phenocopies wildtype and phoshonull mutant phenocopies CDKi inhibition in terms of GFP-RECQL4 localization to laser-induced DNA damage. However, to demonstrate that it is the CDK-mediated phosphorylation of RECQL4 that promotes its recruitment to DNA damage, the authors should demonstrate that phosphomimetic mutant RECQL4-2D can rescue the localization defect rendered by the inhibition of CDKs. Furthermore, the authors demonstrate in Fig 3H that specific inhibition of CDK diminished the interaction of RECQL4 and MRE11, thus suggesting CDK-mediated phosphorylation of RECQL4 is required for its interaction with MRE11, the authors should include results from IP of RECQL4 phosphonull and phosphomimetic mutant to verify that direct phosphorylation of RECQL4 is required for its interaction with MRE11.

Response: This is a very good point.

As suggested, we performed several new experiments to address these questions. First, to investigate whether the phosphorylation-mimetic mutant RECQL4-2D can rescue the recruitment defect by CDK1 and CDK2 inhibitors, we performed a new laser-treated recruitment of RECQL4-2D, RECQ4-2A, and WT RECQL4, in the presence or absence of these inhibitors. The new data is now included as Fig. 3e. Consistent with the findings in the previous version of the manuscript, GFP-tagged RECQL4Wt and RECQL4-2D are rapidly recruited to laser-induced DSBs but the extent of recruitment of RECQL4-2A is repressed, indicating that phosphorylation of RECQL4 on Ser89 and Ser251 promotes RECQL4 recruitment to DSBs. With the treatment of CDK1 and CDK2 inhibitors, the extent of recruitment of RECQL4Wt and RECQL4-2D to DSB sites was suppressed. Inhibition of CDK1 and CDK2 has a wide effect in cells, and the upstream factors could be dysfunctional after CDKs inhibition. Previously, we reported that recruitment of RECQL4 depends on MRE11 (Lu et al., Cell Reports, 2016, 16:161-173), and we therefore wanted to determine whether recruitment of MRE11 was inhibited by CDK inhibitors, which was also a question raised by Reviewer 2. We compared the recruitment dynamics of YFP-tagged MRE11 in the absence and presence of CDK1 and CDK2 inhibitors. We found that treatment of CDK1 and CDK2 inhibitors significantly suppressed the enrichment of MRE11 at DSB sites. This is now shown in the new Fig. 3f. Analysis of the amount of chromatin-bound MRE11 also supports these findings, since we found that accumulation of MRE11 to chromatin was repressed in U2OS cells treated with CDK1 and CDK2 inhibitors (Supplemental fig. 3g). Combined, these findings explain why the CDK1 and CDK2 inhibitors repress the recruitment of phosphorylation-mimetic mutant RECQL4-2D.

In the past version of this manuscript, we showed that CDK2 inhibition disturbs the interaction between MRE11 and GFP-tagged RECQL4. As suggested by Reviewer 2, we have now also measured the

MRE11/RECQL4 interaction using endogenous proteins, and tested whether CDK inhibition reduced this physical interaction. As showed in Fig. 3g, RECQL4 forms a complex with MRE11, and this interaction was disrupted by inhibition of CDK1 and CDK2.

To test whether phosphorylation of RECQL4 is required for interaction with MRE11, we added another IP to exam the interaction between MRE11 and RECQL4Wt, RECQL4-2A or RECQL4-2D, as suggested. We pulled down GFP-tagged RECQL4Wt, RECQL4-2A and RECQL4-2D as well as GFP from the irradiated cells, and detected MRE11 and Ku70, which was also suggested by Reviewer 2. As shown in Fig. 3h, RECQL4-2A pulled down much less MRE11 but more Ku70 than RECQL4Wt and RECQL4-2D, indicating that CDK-dependent phosphorylation on Ser89 and Ser251 regulates the interaction of RECQL4 with MRE11 and Ku70.

Please see new data in Fig. 3e, 3f, 3g, and 3h, as well as Supplemental Fig. 3g. And the related new text are from Page 10 to 13.

Figure 4: Here authors intend to demonstrate that the E3 ubiquitin ligase complex that found to interact with RECQL4, shown by IP-MS, also promotes the ubiquitylation of RECQL4 and that this ubiquitylation is dependent on phosphorylation. Furthermore, the ubiquitylation of RECQL4 is also required for its localization to damaged chromatin. The authors demonstrated the requirement of phosphorylation for ubiquitylation through either siRNA-mediated depletion of E3 ubiquitylating enzyme complex, or expression of phosphonull form of RECQL4 that resulted in diminishing level of ubiquitylation by immunoblotting, compared to expression of wildtype. However, for a more rigorous and specific proof, the authors should purify WT and phosphonull mutant and detect ubiquitylation sites, ie putative lysine residues, by MS/MS, and subsequently express ubiquitylating-null mutant to look for resulting defect in RECQL4 localization in response to laser induced damage.

Response: To address this concern, we purified the 3X FLAG-tagged RECQL4Wt and RECQL4-2A, and submitted the proteins to Harvard Taplin Mass Spectro Center to identify the ubiquitination sites. Although mass spectrometry-based proteomics has advanced significantly in the past decade, the identification of ubiquitination sites by mass spectrometry on a target protein is still challenging (Zhu et al, Biochemistry and Biophysics Reports, 2016, 430-438). Unfortunately, we did not succeed in the identification of ubiquitination sites on RECQL4 and its mutant by mass spectrometry. Since RECQL4 has more than 30 lysine residues, it is technically challenging to mutate these residues and identify specific residues.

Although we are unable to identify the ubiquitination sites on RECQL4, our data clearly demonstrate that ubiquitination of RECQL4 depends on DDB1-CUL4A E3 ubiquitin ligase, and that DDB1 promotes the recruitment of RECQL4 to laser-induced DSBs. To further support this finding, we have now added three additional experiments in the revised MS. As suggested by Reviewer 2, we now show that ubiquitination of endogenous RECQL4 depends on the DDB1-CUL4A complex (Fig.4c), and that inhibition of CDK1 and CDK2 reduces the ubiquitination level of endogenous RECQL4 (Fig.4d). To strengthen the statement that recruitment of RECQL4 depends on the DDB1-CUL4A complex, we now show that disruption of the DDB1-CUL4A complex by siRNA-mediated knockdown of CUL4A reduces ubiquitination of endogenous RECQL4 (Fig. 4c) and also represses accumulation of RECQL4 on chromatin after IR treatment (Fig. 4h).

As suggested by Reviewer 2, we investigated whether less ubiquitination of RECQL4-2A is caused by a weak interaction between RECQL4 and the E3 ubiquitin ligase. We found that RECQL4-2A has lower affinity for DDB1, compared to RECQL4Wt and RECQL4-2D (Fig. 4f). This physical interaction also supports our claim of DDB1-dependent ubiquitination of RECQL4.

Taken together, we believe that we have demonstrated that DDB1-CUL4A-mediated ubiquitination of RECQL4 promotes its recruitment to DSBs.

Please see new data in Fig. 4c, 4d, 4f and 4h, as well as the related text in Pages 14 and 15.

Figure 5: The authors showed that phosphorylation of RECQL4 stimulates its enzymatic activities on DNA, including helicase and annealing activities. The authors demonstrated the helicase activities and RECQL4's ability to bind DNA using splayed-arm/Y-structure duplex DNA oligo structure. I think the authors could better substantiate their conclusions regarding RECQL4 stimulating end resection with MRE11 during S/G2 by demonstrating that RECQL4 display substrate binding and helicase activity preference specifically towards duplex with only 3' ssDNA overhang, which better mimics physiological substrate for end resection; as oppose to Y structure, which supports RECQL4 function in the context of replication. Testing alternative substrate to identify preferred substrate of RECQL4 could also render more obvious DNA binding product that that shown in Fig 5C.

Response: Paull and her colleagues showed that MRN can unwind 15-20 base pairs at the end of the duplex, and hold the branched structure open for minutes, which promotes recruitment of downstream enzymes to break sites and signaling activities (Cannon et al, PNAS, 2013, 110:18868-18873). RECQL4 is recruited by MRN, before it promotes CtIP recruitment for initial 5' end resection (Lu et al., Cell Reports, 2016, 16:161-173). Thus, short 3' ssDNA tails generated after MRN/CtIP-mediated initial resection depends on RECQL4. Therefore, we believe that a Y-structured duplex DNA substrate is a good mimicking substrate for testing the DNA unwinding activity and DNA binding activity of RECQL4.

For the reviewer concern, we performed gel mobility shift assay to assess DNA binding ability of these proteins to a duplex with 3' ssDNA overhang, and found that the three proteins showed similar affinity to the substrate (Supplemental fig. 5b). These findings suggest that ser89/Ser251 phosphorylation does not alter DNA binding of RECQL4.

Please see new data in Supplemental fig. 5b, and the blue text in Pages 16 and 17, as well as Page 23 in the section of Discussion.

Figure 6: Here the authors demonstrates phosphorylation of REQL4 is required for end resection, thereby HR efficiency. Although the results for WT and phosphomimetic mutant are clear compare to phosnull mutant (Fig 6 A, B, C, D), it would be important for authors to include MRE11 depletion as a control for these experiments to compare the level of end resection defect manifested by phosnull mutant since one of authors key points is that phosphorylation of RECQL4 is required for MRE11 interaction hence phosnull RECQL4 should phenocopy MRE11 depletion.

Response: As we reported previously, depletion of RECQL4 or MRE11 causes loss of DNA resection or HR to a similar extent as does double knockdown of RECQL4 and MRE11 (Lu, et al, Cell Reports, 2016). MRE11 and RECQL4 work together in DNA end resection, therefore a 5' end resection assay would be best for showing the direct consequence of a dysfunctional RECQL4/MRE11 interaction due to Ser89/Ser251 mutations. Thus, as suggested, we re-performed the DNA resection assay and included a MRE11 knockdown. We also included analysis at multiple sites to show resection length in vivo. Knockdown of RECQL4 or MRE11 resulted in a severe loss of ssDNA content generated by 5' end resection, (Fig. 6a, and Supplemental fig. 6a). And expression of RECQL4 Wt or RECQL4-2D can rescue the loss of DNA resection in siRQ4-treated cells. RECQL4-2A only restored the 5' end resection partially, much less than that by RECQL4 Wt or RECQL4-2D. Therefore, Ser89/Ser251 phosphorylation is important for RECQL4 to function in MRE-mediated DNA end resection. Please see new Fig. 6a, and Supplemental Fig.6a.

Overall, the authors showcased results that demonstrated important regulatory mechanisms, through post-translational modification of RECQL4 helicase, that promotes HR during S/G2. If the authors could include results from suggested experiments mentioned herein, the conclusion would be better supported.

Response: Thank you very much for these comments and the new experiments have definitely improved this manuscript.

Reviewer #2 (Remarks to the Author):

In their manuscript entitled 'Cell cycle-dependent phosphorylation and ubiquitination regulates RECQL4 pathway choice in DNA double-strand break repair', Lu et al. describe the role of RECQL4 phosphorylation and ubiquitylation in cell cycle regulation of double strand break repair. The authors claim that CDK-induced phosphorylation of RECQL4, which is followed by ubiquitination by CUL4-complexes, favors binding to MRE11 hence stimulating DNA end resection in S-phase, while in G1 phase, RECQL4 binds to the KU complex and stimulates NHEJ. The authors nicely show RECQL4 is phosphorylated by CDK1/2 during S-phase, thereby increasing its binding to MRE11 and recruitment to DSBs. However, they previously already showed that RECQL4 stimulates Mre11-dependent end-resection and homologous recombination, severely limiting the novelty of the current work.

With regard to this latter point, the following major points should be addressed to warrant publication of the manuscript:

1. whether this phosphorylation affects KU binding (and thereby explains the shift in complex formation from G1 to S phase) and NHEJ repair.
2. mechanistic insight into how CDK-mediated phosphorylation drives Mre11-dependent end-resection and DSB repair by homologous recombination
3. what the effect of the CUL4-mediated ubiquitylation is on DSB repair pathway choice.

Response: We thank the reviewer for these comments. We have performed a number of suggested experiments to improve our manuscript.

The paper should be improved by adding the following experiments to test:

- RECQL4-2D mutants for KU binding and DSB recruitment in G1

Response: This is good point. We added two experiments to address this question. First, we did a CoIP to compare the binding ability of RECQL4Wt, RECQL4-2A and RECQL4-2D to MRE11 and Ku70 in irradiated HEK293T cells. As shown in Fig. 3h, GFP-RECQL4-2A pulled down less MRE11 but more Ku70, indicating the regulation of interaction partner of RECQL4 by phosphorylation on Ser89/Ser251. We then measured the chromatin-binding of 3XFLAG-RECQL4-2D in G1 U2OS cells with endogenous RECQL4 depleted. After IR stress, we did find an increase of FLAG-RECQL4-2D on chromatin (Supplemental fig. 3j), indicating that RECQL4-2D can be recruited to DSBs in G1 cells, which is also consistent with the following findings: 1. MRE11 is recruited to DSBs in G1 cells (Kim *et al*, J Cell Biology, 2005, 341–347); 2, RECQL4-2D interacts with Ku70 (Fig. 3h); 3, RECQL4-2D partially restores the NHEJ loss in RQ4-depleted EJ5 U2OS cells (Fig. 6d). Please see the new data in Fig. 3h, Supplemental Fig. 3j, and the related text in Page 13.

- if expression of RECQL4-2D mutant affects NHEJ (?)

Response: As suggested, we added the NHEJ experiment with EJ5 U2OS cells, and found that RECQL4-2D expression can partially restore NHEJ in RQ4-depleted cells (Fig. 6d). See more explanation in the previous question. Please see new data in Fig. 6d, Supplemental fig. 6e and the text in Page 19.

- whether the RECQL4-2A mutant shows reduced interaction with Mre11

Response: In the new experiment, we clearly show that RECQL4-2A has less interaction with MRE11, compared to RECQL4Wt and RECQL4-2D (Fig. 3h). Combined with the finding that CDK1 and CDK2 inhibitors repress RECQL4-MRE11 interaction in U2OS cells (Fig. 3g), this indicates that Ser89/Ser251 phosphorylation by CDK is important for RECQL4 to interact with MRE11. Please see the new data in Fig. 3h and the related text in Page 13.

- whether endogenous RECQL4 and Mre11 interact in reciprocal IPs and whether inhibition of CDK1/2 affects this interaction, since all experiments were done following overexpression of FLAG-tagged RECQL4

Response: As suggested, we performed a CoIP to measure the interaction between endogenous RECQL4 and MRE11. Using MRE11 antibody, we were able to pull down RECQL4 with MRE11 from irradiated U2OS cells (Fig. 3g), indicating that these two proteins interact in cells. In the same experiment, we also tested the interaction after treatment of CDK1 and CDK2 inhibitors. Consistent with the data in Fig. 3f, the interaction between MRE11 and RECQL4 is disrupted by CDK1 and CDK2 inhibition (Fig. 3g), indicating CDK1 and CDK2 regulate MRE11/RECQL4 interaction. Please see the new data in Fig. 3g, and the related text in Page 12.

- whether inhibition of CDK1/2 also affect phosphorylation and damage recruitment of endogenous RECQL4

Response: We pulled down endogenous RECQL4 from U2OS cells, after treatment with a pan-CDK inhibitor roscovitine, selective CDK2 inhibitors or DMSO. Treatment with roscovitine or CDK2 inhibitor reduced phosphorylation by 60% on the SP motif in RECQL4 (Supplemental fig. 3b). Consistent with this

finding, dynamic recruitment assays showed that roscovitine or CDK2 inhibitor treatment significantly repressed GFP-RECQL4 recruitment (Supplemental fig 3e and Supplemental fig. 3f).

As suggested, we also investigated endogenous RECQL4 recruitment to DNA damage by measuring accumulation of RECQL4 to chromatin after 10Gy IR treatment. We found that IR treatment causes RECQL4 to accumulate in the chromatin fraction of U2OS cells, and inhibition of CDK1 and CDK2 inhibits RECQL4's accumulation on chromatin after DNA damage (Supplemental fig. 3g).

These findings confirm our findings that CDK1/2-mediated phosphorylation promotes RECQL4 recruitment to DSBs.

Please see the new data in Supplemental Fig. 3b and 3g, and the related text in Pages 10 and 11.

- whether Mre11 is still recruited following CDK1/2-inhibition

Response: This is a good point. As suggested by Reviewer 1, we have now analyzed the recruitment of the phosphorylation-mimic mutant RECQL4-2D in response to DSBs in the presence of CDK inhibitors, and found that it is still repressed (Fig. 3e). However, we reasoned that CDK 1 and CDK 2 inhibition could also target upstream factors and lead to RECQL4's failure to localize to DNA damage. Considering that MRE11 regulates the recruitment of RECQL4 to DSB, we analyzed whether MRE11 was recruited in response to DNA damage after CDK1 and CDK2 inhibition. We assessed the dynamics of localization of YFP-MRE11 in U2OS cells with or without CDK1 and CDK2 inhibitors. Interestingly, we found that inhibition of CDK1 and CDK2 greatly attenuates YFP-MRE11 accumulation at laser-induced DSBs (Fig. 3f). By evaluating the endogenous MRE11 content in the chromatin fraction, we showed that CDK1 and CDK2 inhibition reduces accumulation of MRE11 to chromatin in U2OS cells after a 10Gy gamma radiation (Supplemental fig. 3g). Thus, these data support a clear conclusion that CDK1 and CDK2 also regulates recruitment of MRE11 in DSB repair, which also explains why the phosphorylation-mimetic mutant RECQL4-2D is not recruited to DSB sites in the presence of CDK1 and CDK2 inhibitors. Please see new data in Fig. 3e, 3f, and Supplemental Fig. 3g, as well as the related new text from Pages 11 and 12.

- if RECQL4-2D stimulates MRE11 activity in vitro

R: As suggested, we measured the effect of RECQL4-2D on the nuclease activity of MRE11-RAD50-NBS1 on close circle phi X174 ssDNA. RECQL4-2D did not much affect the cleavage efficiency of MRN on the circular ssDNA (Supplemental fig. 5c), indicating that this phosphorylation may not affect RECQL4's stimulation on nuclease activity of MRN in vitro. Please see new data in Supplemental fig. 5c and the changed text in Page 17.

- if endogenous RECQL4 is ubiquitinated by DDB1

Response: Our data shows an interaction of RECQL4 with DDB1 and CUL4A (Supplemental fig. 4a, Fig. 4a and 4b). DDB1 and CUL4A forms a E3 ubiquitin ligase with RBX1. To better answer this question, we performed IP to purify endogenous RECQL4 from U2OS cells treated with control siRNA, DDB1 siRNA or CUL4A siRNA. Both DDB1 and CUL4A were greatly depleted by siRNA-mediated knockdown (Input of

Fig. 4c). We also found that disruption of DDB1/CUL4A E3 ubiquitin ligase, by knocking down either DDB1 or CUL4A, reduced the ubiquitination of RECQL4 (Fig. 4c), further supporting that ubiquitination of RECQL4 depends on DDB1-CUL4A E3 ubiquitin ligase. Please see the new data Fig. 4c and the related text in Page 14.

- the effect of RECQL4 ubiquitylation on MRE11 and KU binding

Response: The best way to answer the question would be to identify the ubiquitination sites and comparing the mutants with wildtype on the interaction between RECQL4 with MRE11 and Ku. Based on Reviewer 1's suggestion, we tried to identify the ubiquitination sites on RECQL4 for further construction of site-directed mutants. Unfortunately, the mass spectrometry analysis did not give a positive result. Identification of posttranslational modifications like ubiquitination on target proteins is quite challenging. Moreover, RECQL4 has over 30 lysine residues, and we are unable to make site-directed mutants to identify the ubiquitination sites. Please see the details in the Reply to Question 4 of Reviewer 1.

However, we do find that the ubiquitination level of RECQL4-2A is significantly lower than that of RECQL4Wt and we showed that RECQL4-2A has high affinity to Ku70 but weaker binding with Ku70. Please see the related discussion above.

- the effect of RECQL4 ubiquitylation on RECQL4 catalytic activity (experiments of Fig. 5 upon purification of RECQL4 from DDB1 siRNA treated cells, or better test a RECQL4 mutant that cannot be ubiquitylated)

Response: Thank you for the comment. As said above, we are unable to identify the ubiquitination site in RECQL4, but there are close connections between Ser89/Ser251 phosphorylation and the ubiquitination level of RECQL4 (Fig. 4d), and we compared the biochemical activities of RECQL4Wt and RECQL4-2A and identified significant difference between two proteins with respect to DNA unwinding activity and strand annealing efficiency at the initial stage, but no significant difference on DNA binding in vitro (Fig. 5 and Supplemental Fig. 5b).

- if the RECQL4-2A mutant shows a reduced DDB1 interaction, to show how RECQL4 phosphorylation could lead to DDB1-dependent ubiquitylation

Response: In the previous version of the manuscript, we reported DDB1-dependent ubiquitination of RECQL4. As suggested, we have now added some experiments to verify this finding. We show here that ubiquitination of endogenous RECQL4 decreases after disruption of DDB1-CUL4A E3 ubiquitin ligase by depleting either DDB1 or CUL4A in U2OS cells. RECQL4-2A shows reduced ubiquitination, as compared to wildtype. In the revised manuscript, a new CoIP shows that RECQL4-2A has less affinity to DDB1 compared to Wildtype RECQL4 and the phosphorylation-mimetic mutant RECQL4-2D (Fig. 4f), suggesting that blocking phosphorylation on Ser89/Ser251 on RECQL4 inhibits the interaction between RECQL4 and DDB1. This explains why ubiquitination of RECQL4-2A is much lower than that of RECQL4Wt. Please see new data in Fig. 4f and the related new text in Page 15.

Minor points:

- At page 6 of the manuscript, first paragraph, the authors write 'which is consistent with previous reports' (the DR-GFP data). Since they are self-citing their own paper, which shows the same results, using the

same assay, I don't find it a strong argument for consistency.

Response: We previously showed that RECQL4 knockdown efficiency affects the extent of HR loss (Lu et al, Cell Reports, 2016, 16:161-173). In that paper, over 90% of RECQL4 was depleted by RQ4 siRNA, which gave relative HR levels of $26.6\% \pm 1.8\%$. Here, 80% knockdown of RECQL4 gives $36.6\% \pm 7.9\%$ of HR (Fig. 1a). We consider this a reasonable variation for biological repeats and this experiment also verifies the conclusion that RECQL4 play an important role in HR. Please see the changed text in Page 6.

- Figure 1 seems to contain data that have already been reported before by the authors.

Response: We have identified the role of RECQL4 in NHEJ and HR (Shamanna et al, Carcinogenesis, 2014, 35: 2415-2424; Lu et al, Cell Reports, 2016, 16:161-173). Here we report how one protein participates in two different DSB repair pathways, which is rare but important and novel. To help readers understand and follow the flow of the story, we independently performed the HR assay using DR-GFP cells as explained above. For measuring NHEJ efficiency, we used a new reporter cell line EJ5 U2OS from Dr. Jeremy Stark (Bennardo et al PloS Genetics, 2008, e1000110), and we found a similar phenomenon as was previously reported (Shamanna et al, Carcinogenesis, 2014, 35:2415-2424). Therefore, we would like to keep the data in and we replace the word "demonstrate" with "verify" to indicate it. Please see the changed text in Page 6.

- Given the role of RECQL4 on DNA replication, does RECQL4 knock-down or knock-out affect cell cycle, thereby affecting HR rate?

Response: According to our previous finding (Lu et al, Cell Reports, 2016, 16:161-173) and other findings (Park et al, DNA Cell Biol, 2006, 25: 696-703), knockdown of RECQL4 does not significantly affect cell cycle progression in U2OS cells and 293T cells. Presumably the residual RECQL4 is enough for cells to replicate. Therefore, we consider that loss of HR repair in RECQL4-depleted cells is not a result of cell cycle failure but is caused by limited involvement of RECQL4 in HR.

- Figure 1A-B, the effect of RECQL4 knockdown on HR and NHEJ is comparable, while the knockdown is not (poor knockdown in A, highly efficient knockdown in B). How can this be explained?

Response: Actually, the knockdown efficiency of RECQL4 clearly affects the extent of HR or NHEJ loss in the GFP-based reporter systems. For HR, we previously described the phenomenon (Lu et al, Cell Reports, 2016, 16:161-173). For NHEJ, we measured the rejoining efficiency with a new reporter system, EJ-5 U2OS cells, a gift from Dr. Jeremy Stark. The result in Fig. 1b is from three independent experiments. Due to a fluctuation of siRNA-mediated knockdown efficiency, the NHEJ efficiency accordingly changes. Thus although the error bar is big, the results are still significantly different between control and RECQL4-depleted cells. The blot shown is one of the three repeats.

- It is unclear to me whether the RECQL4 recruitment to laser tracks experiments in figure 3f/g and 4g are based on S-phase cells only. If not, I wonder whether the data is skewed by cells in G1 binding to DSBs via KU.

Response: We did not synchronize the cell cycle to a specific phase for analyzing real-time recruitment of RECQL4. The cells we collected were randomly picked up from the view of microscope, and then stripped with laser. The data were calculated from a number of cells. We show here that the treatment of CDK1 and CDK2 inhibitors or substitution of Ser89/Ser251 repressed the recruitment of RECQL4 (Fig. 3e). It is possible that some of cells from G1 may alter the intensity of the fluorescence. However, even including the data from these G1 cells, there is still a big difference between CDK inhibitor-treated and control cells. Therefore, the presented data clearly supports our hypothesis.

- Figure 4e, confirm the effect of RECQL4 phosphorylation on ubiquitylation by testing ubiquitylation of RECQL4 after CDK1/2 inhibition.

Response: Upon this suggestion, we added a new IP, where we used the RECQL4 antibody to purify endogenous RECQL4 from denatured cell lysate of DMSO or CDK1/ CDK2 inhibitor-treated U2OS cells, and detected the ubiquitination level with an anti-Ub antibody. The result shows that CDK1 and CDK2 inhibition greatly reduces the ubiquitination level of endogenous RECQL4 (Fig. 4d), which supports the finding that the phosphorylation-null mutant RECQL4-2A has less ubiquitination. Please see the new data in Fig. 4d, and the changed text in Page 15.

- In the discussion, the authors suggest that DDB1/CUL4-mediated ubiquitylation of RECQL4 depends on the cell cycle since it depends on the cell cycle regulated phosphorylation. It would be of interest to confirm this hypothesis.

Response: Ser89/S251 phosphorylation required for DDB1-CUL4A mediated ubiquitination of RECQL4 (Fig. 4), and phosphorylation by CDK1/2 is cell cycle regulated. Therefore, we assume that ubiquitination of RECQL4 is cell cycle regulated, but we state in the text that this needs further substantiation going forward.

Reviewers' Comments:

Reviewer #2 (Remarks to the Author)

I appreciate the effort of the authors to respond to all issues and perform a major revision. While several issues have been addressed satisfactorily, my overall impression is that the newly added data do not substantially strengthen the major conclusions of the manuscript. Some major and minor issues remain, which are listed below.

Major issues:

- It is still unclear what the role of RECQL4 ubiquitylation is in regulating RECQL4 function during HR at the molecular level. It is understandable that the mapping of ubiquitylation sites is challenging, yet such an undertaking would be required to construct a ubiquitylation-dead version of RECQL4. Studying such a mutant would be the way to obtain insight into the ubiquitylation-dependent regulation of RECQL4 during HR. The lack of such data questions the additive value of the ubiquitylation data in the current manuscript.
- The GFP reporter studies in figure 6c and 6d provide important functional data on RECQL4's role in regulating pathway choice. However, the data are not very convincing as the effect of RECQL4-2A on DDB1-mediated ubiquitylation (Figure 4e), DDB1 binding (Figure 4f) and HR (Figure 6c) is very mild, while that on NHEJ, although being elevated (6d), is not significant. This weakens the major conclusion on an important role for RECQL4 phosphorylation/ubiquitylation in regulating pathway choice.

Minor issues:

- Abstract line 19-20: remove "to ensure"
- Abstract line 33-34: "Collectively, we identified how RECQL4 modulates DSB repair pathway choice by differentially regulating NHEJ and HR in a cell cycle dependent manner." Not sure if I agree with this conclusion. I think a mechanism for the regulation of RECQL4 during HR, rather than NHEJ, was identified. This is also what the authors conclude themselves at the end of the introduction (lines 88-94). Please change accordingly.

REVIEWERS' COMMENTS:

Reviewer #2 (Remarks to the Author):

I appreciate the effort of the authors to respond to all issues and perform a major revision. While several issues have been addressed satisfactorily, my overall impression is that the newly added data do not substantially strengthen the major conclusions of the manuscript. Some major and minor issues remain, which are listed below.

Major issues:

- It is still unclear what the role of RECQL4 ubiquitylation is in regulating RECQL4 function during HR at the molecular level. It is understandable that the mapping of ubiquitylation sites is challenging, yet such an undertaking would be required to construct a ubiquitylation-dead version of RECQL4. Studying such a mutant would be the way to obtain insight into the ubiquitylation-dependent regulation of RECQL4 during HR. The lack of such data questions the additive value of the ubiquitylation data in the current manuscript.

Response: We completely agree with the Reviewer that identification and characterization of ubiquitylation sites on RECQL4 would provide insight of the molecular mechanisms underlying the stimulation of DDB1-CUL4A-mediated ubiquitination on RECQL4 in DSB repair. Unfortunately, we failed to identify the sites due to technical limitations. However, we provide clear data that ubiquitination of RECQL4 is mediated by the DDB1-CUL4A E3 ubiquitin ligase, which follows CDK-mediated phosphorylation. Moreover, depletion of either DDB1 or CUL4A greatly suppresses recruitment of RECQL4 to DSBs, indicating the biological function of DDB1-CUL4A-mediated ubiquitination of RECQL4. Thus, we believe that the data strengthen the insight mechanisms of CDK-mediated phosphorylation of RECQL4 in DSB repair. However, since the data does not provide complete direct evidence on cell cycle dependent ubiquitination of RECQL4, and in response to the comments made by this reviewer, we have changed the Title to better reflect the totality of our study. Please see the changes in the title in Page 1.

- The GFP reporter studies in figure 6c and 6d provide important functional data on RECQL4's role in regulating pathway choice. However, the data are not very convincing as the effect of RECQL4-2A on DDB1-mediated ubiquitylation (Figure 4e), DDB1 binding (Figure 4f) and HR (Figure 6c) is very mild, while that on NHEJ, although being elevated (6d), is not significant. This weakens the major conclusion on an important role for RECQL4 phosphorylation/ubiquitylation in regulating pathway choice.

Response: We agree with the Reviewer that the decrease in HR efficiency in the GFP reporter assays is not drastic in the 2A complemented cells (Figure 6c), but the decrease is significant compared to WT. Also, it should be noted that one issue with the GFP reporter assay is that it observes completion of repair over a long time frame and only shows terminal events. Due to this issue, we used other assays to assess the onset of HR including a DNA end resection assay (Figure 6a) and RPA focus formation following DNA damage (Figure 6b). These assays, along with the GFP reporter assay, show that the cells expressing the 2A protein have significantly less DNA end resection and HR compared to WT, supporting our hypothesis that phosphorylation of RECQL4 is important for promoting HR.

Minor issues:

- Abstract line 19-20: remove “to ensure”.

Response: We have edited the abstract.

- Abstract line 33-34: “Collectively, we identified how RECQL4 modulates DSB repair pathway choice by differentially regulating NHEJ and HR in a cell cycle dependent manner.” Not sure if I agree with this conclusion. I think a mechanism for the regulation of RECQL4 during HR, rather than NHEJ, was identified. This is also what the authors conclude themselves at the end of the introduction (lines 88-94). Please change accordingly.

Response: We have changed it as “Collectively, we report that RECQL4 modulates DSB repair pathway choice by differentially regulating NHEJ and HR in a cell cycle dependent manner.”